# Cyclin A triggers Mitosis either via the Greatwall kinase pathway or Cyclin B

Nadia Hégarat[1],[†], Adrijana Crncec[1],[†] (ID), Maria F Suarez Peredo Rodriguez[1] (ID), Fabio Echegaray Iturra[1] (ID), Yan Gu[1], Oliver Busby[1], Paul F Lang[2], Alexis R Barr[3],[4] (ID), Chris Bakal[5], Masato T Kanemaki[6],[7], Angus I Lamond[8] (ID), Bela Novak[2] (ID), Tony Ly[9],* (ID) & Helfrid Hochegger[1],** (ID)

## Abstract

Two mitotic cyclin types, cyclin A and B, exist in higher eukaryotes, but their specialised functions in mitosis are incompletely understood. Using degron tags for rapid inducible protein removal, we analyse how acute depletion of these proteins affects mitosis. Loss of cyclin A in G2-phase prevents mitotic entry. Cells lacking cyclin B can enter mitosis and phosphorylate most mitotic proteins, because of parallel PP2A:B55 phosphatase inactivation by Greatwall kinase. The final barrier to mitotic establishment corresponds to nuclear envelope breakdown, which requires a decisive shift in the balance of cyclin-dependent kinase Cdk1 and PP2A:B55 activity. Beyond this point, cyclin B/Cdk1 is essential for phosphorylation of a distinct subset of mitotic Cdk1 substrates that are essential to complete cell division. Our results identify how cyclin A, cyclin B and Greatwall kinase coordinate mitotic progression by increasing levels of Cdk1-dependent substrate phosphorylation.

**Keywords** Cdk1; Cyclin; Greatwall; MASTL; PP2A
**Subject Categories** Cell Cycle
**EMBO Journal (2020) 39: e104419**

## Introduction

Cdk1 phosphorylates over 1,000 proteins (Daub *et al*, 2008; Dephoure *et al*, 2008) within the brief 20–30 min window of mitotic entry. Rising levels of Cdk1 activity correlate with centrosome separation and chromosome condensation in prophase, followed by nuclear envelope breakdown (NEBD) and mitotic spindle formation in prometaphase, and the alignment of bi-oriented sister chromatids at the metaphase plate (Gavet & Pines, 2010). Binding of a cyclin partner is critical for allosteric activation of Cdks. Two families of mitotic cyclins, termed A and B, work with Cdk1 to orchestrate mitotic entry in higher eukaryotes (Fung & Poon, 2005; Hochegger *et al*, 2008). Despite the central importance of these proteins for cell cycle control, the functional specialisation of mammalian A- and B-type cyclins remains unclear. Following the depletion of maternal pools of early embryonic cyclins A1 and B3, somatic mammalian cells express one A-type cyclin, A2, and two B-type cyclins, B1 and B2. Specific binding partners for cyclin A2 and B1 have been identified (Pagliuca *et al*, 2011), but the precise contributions of these two cyclins to mitotic entry remain unclear.

Genetic depletion in mice suggests an essential role for cyclin A2 for early development, but not for embryonic fibroblast and liver cell proliferation (Kalaszczynska *et al*, 2009; Gopinathan *et al*, 2014). Antibody injections against cyclin A2 or depletion of human cyclin A2 by siRNA delay mitotic entry, and this is further enhanced by co-depletion of cyclin B1 (Furuno *et al*, 1999; Fung *et al*, 2007; Gong *et al*, 2007; Gong & Ferrell, 2010). A mechanism involving Plk1 activation has been suggested (Gheghiani *et al*, 2017; Vigneron *et al*, 2018), although an essential role of Plk1 in the G2/M transition remains contentious (Burkard *et al*, 2007). Likewise, work in mammalian cell extracts documented how cyclin A synergises with cyclin B to control the mitotic entry threshold at the level for Cdk1 activation (Deibler & Kirschner, 2010). A confounding factor in the genetic analysis of cyclin A2 has been its dual role in S phase and mitosis (Katsuno *et al*, 2009; Chibazakura *et al*, 2011), making it difficult to directly investigate G2-specific defects. Furthermore, mitotic-specific functions of cyclin A2 in modulating Plk1 activity have also been reported (Kabeche & Compton, 2013; Dumitru *et al*, 2017).

1 Genome Damage and Stability Centre, School of Life Sciences, University of Sussex, Brighton, UK
2 Department of Biochemistry, University of Oxford, Oxford, UK
3 MRC London Institute of Medical Science, Imperial College, London, UK
4 Institute of Clinical Sciences, Faculty of Medicine, Imperial College, London, UK
5 Institute for Cancer Research, Chester Beatty Laboratories, London, UK
6 National Institute of Genetics, Research Organization of Information and Systems (ROIS), Mishima, Japan
7 Department of Genetics, SOKENDAI (The Graduate University of Advanced Studies), Mishima, Japan
8 Centre for Gene Regulation and Expression, School of Life Sciences, University of Dundee, Dundee, UK
9 Wellcome Trust Centre for Cell Biology, University of Edinburgh, Edinburgh, UK
*Corresponding author. Tel: +44 131 650 7106; E-mail: tly@ed.ac.uk
**Corresponding author. Tel: +44 1273 877510; E-mail: hh65@sussex.ac.uk
†These authors contributed equally to this work

Murine cyclin B1 is essential for development (Brandeis *et al*, 1998) and critical for mitotic entry in early mouse embryos (Strauss *et al*, 2018). Conversely, mice lacking cyclin B2 live healthily, without apparent defects (Brandeis *et al*, 1998). These results stand in stark contrast to observations from experiments involving siRNA depletion of B-type cyclins in human somatic cell lines that show surprisingly mild mitotic entry defects (Chen *et al*, 2008; Gong & Ferrell, 2010; Soni *et al*, 2014). The observed unperturbed G2/M transition in cyclin B-depleted human cells could be explained by a synergy between A- and B-type cyclins in mitotic entry (Gong & Ferrell, 2010). Cyclin B could also be critical to elevate Cdk1 activity after mitotic entry (Lindqvist *et al*, 2007) and compensate for the premature loss of cyclin A (Geley *et al*, 2001; Voets *et al*, 2015). Specific localisation of cyclin B in mitosis may also contribute to its specialised functions, as was recently shown in connection with Mps1 activation and spindle assembly checkpoint (SAC) signalling (Alfonso-Pérez *et al*, 2019; Hayward *et al*, 2019).

In parallel to mitotic cyclin/Cdk1 activation, the inactivation of the Cdk1 counteracting phosphatase PP2A:B55 by Greatwall kinase (GWL) via its substrates ENSA and ARPP19 also plays a critical role in mitotic entry in *Xenopus* egg extracts (Castilho *et al*, 2009; Gharbi-Ayachi *et al*, 2010; Mochida *et al*, 2010). However, human and mouse cells lacking Greatwall kinase readily enter mitosis (Burgess *et al*, 2010; Alvarez-Fernández *et al*, 2013), but fail to undergo accurate sister-chromatid segregation and cell division (Cundell *et al*, 2016). Thus, it appears that human cells can enter mitosis in the absence of cyclin B or Greatwall, but it remains unclear to what extent the kinase activation and phosphatase inactivation pathways compensate for each other. Moreover, the precise G2-specific role of cyclin A in this regulatory network is still an open question. In this study, we have taken a genetic approach based on optimised degron tagging to address these questions.

# Results

## A double degron of mAID and SMASh allows efficient depletion of cyclin A and B in RPE-1 cells

Rapid, precise and uniform depletion is critical to investigate the functions of mitotic cyclins within a single cell cycle. We have recently reported that a combination of mAID- (Natsume *et al*, 2016) and SMASh- (Chung *et al*, 2015) tags results in highly efficient induced degradation (Lemmens *et al*, 2018), and applied this strategy (see Fig 1A) to comprehensively analyse the functions of cyclin A2 and B1 in a non-transformed, h-TERT immortalised human epithelial cell line, RPE-1. Firstly, we designed gene targeting constructs and gRNA/Cas9 constructs to inducibly express OsTIR1 from the human Rosa26 locus (Fig 1B, upper panel). This was critical to achieve a controllable and uniform degradation of mAID-tagged proteins. Next, we introduced the mAID-SMASh double tag in the C terminus of the human CCNA2 and CCNB1 genes using targeting vectors and gRNA/Cas9 cleavage (See Fig 1B middle and bottom panel and Materials and Methods). Targeted clones were selected using a T2A coupled gene conferring resistance to Neomycin. Gene targeting was confirmed by genomic PCR (Fig 1C). Both cyclins were essentially undetectable by immunoblotting after 2–4 h of both mAID- and SMASh-induced degradation (Fig 1D) using the "DIA" cocktail (Doxycycline/IAA for mAID and Asv for SMASh, see Materials and Methods section). Quantification of this depletion experiment (Fig 1E) shows 50% depletion within the first hour and an ultimate depletion to < 5% of control protein levels within 2–4 h. Activation of both degradation systems was highly effective to trigger rapid degradation of either cyclin compared to the individual degron systems. We observed a strong synergy in the combined degron compared to inducing the individual degradation systems (Fig 1F). Further analysis confirmed that depletion of cyclin

---

**Figure 1. Generation and characterisation of cell lines expressing mAID-SMASh-tagged cyclin A2 or B1.**

A   Schematic of protein degradation using AID and SMASh systems. The SMASh tag consists of the self-cleaving viral HCV NS3 protease fused in cis to a constitutive destabilising peptide. The tag detaches itself from the target protein in the off condition. Upon inhibition by the protease inhibitor, asunaprevir (Asv), the tag remains on the target protein and exposes it to the cellular degradation machinery, while the old protein is depleted depending on its specific half-life. The AID-tag relies on the expression of the plant F-box protein TIR1 that combines with mammalian Skp1 and Cul1 to form a functional SCF E3 ligase that is activated to ubiquitylate the AID degron tag upon stimulation by the hormone Auxin (IAA). In the case of the double tag combination, newly synthesised proteins are depleted by the two independent degradation systems. Old proteins are destabilised by the mAID-tag (mini-AID).

B   Knock-in strategy for inducible OsTIR1 expression at the human Rosa26 locus and endogenous cyclin A2 and B1 protein tagging. We integrated mAID-SMASh-T2A-neomycin in-frame at the C terminus of the coding sequence of cyclins A2 and B1. The chosen length for the homologous regions for gene targeting was ~ 1 kb.

C   To verify gene targeting primers were designed in the genomic locus and in the cassette (indicated in (B)) and used to amplify genomic DNA from different cell lines: wild-type RPE-1, RPE-1 expressing OsTIR1, CycA2[dd], CycB1[dd] and primers amplifying a region of the Kif23 gene were used as a positive control (see Material and Methods section for primer sequences).

D, E   (D) Representative immunoblots and (E) quantitation showing timecourse of induced degradation of double degron-tagged cyclin A and cyclin B (A2[dd], B1[dd]). * indicates cross-reacting band. (*n* = 3 experiments, s.d. indicated by error bars). Doxycycline/IAA/Asv (DIA, see Material and Methods) was added to asynchronous cells for indicated time before extract preparation and immunoblotting.

F   Comparison of protein degradation efficiency using different degrons. Cells were incubated for indicated time periods with Asv or DOX/IAA or both (DIA, see Material and Methods) then collected for Western blot analysis. Doxycycline was added 2 h before addition of IAA or Asv/IAA. To determine protein degradation, either cyclin A2 or cyclin B1 was probed as indicated. Anti-myc antibody was used to check OsTIR1-myc expression. Asterisks indicate non-specific protein cross-reacting with the cyclin B1 antibody.

G   Timecourse of DIA-induced cyclin A and B degradation, cross-checking levels of each mitotic cyclin. Treatments were performed as for DIA-treated panels in (E). Cell extracts were checked for both cyclin A2 and B1 levels following DIA addition (* indicates non-specific band).

H   Comparison of DIA and siRNA depletion. Cells were treated with DIA for 4 h or pre-transfected with CycB1 SMARTpool siRNAs 48 h earlier. Cell extracts were analysed by immunoblotting for cyclin B1 (L.E, long exposure, S.E. short exposure) using the same mouse monoclonal antibody as in (E) and (F).

(I)   Assaying cyclin B1 depletion in mitotic cells using a rabbit monoclonal antibody. Extracts from asynchronous or mitotic cells were collected and analysed by immunoblotting using the indicated antibodies.

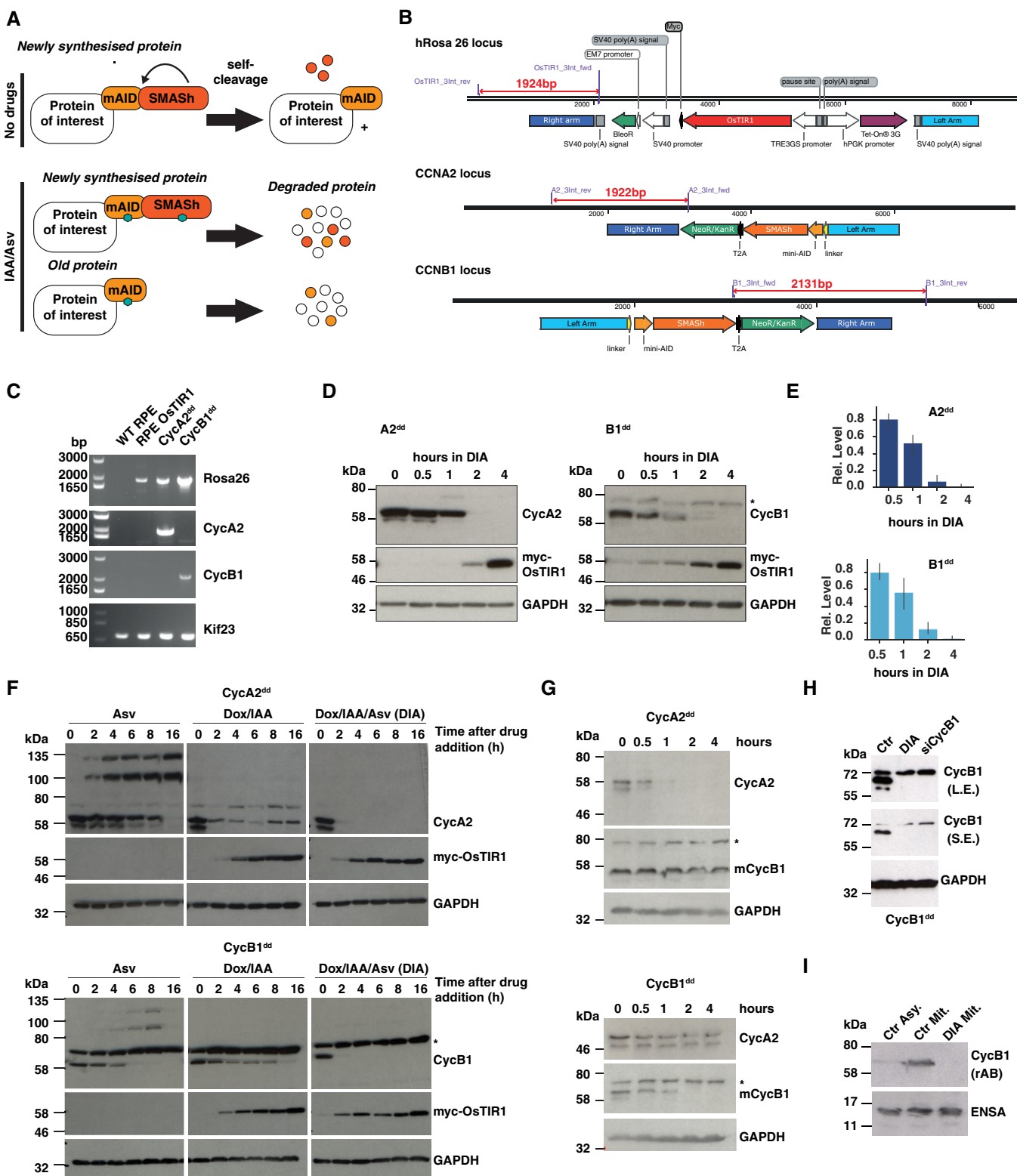

**Figure 1.**

A2 had no impact on cyclin B1 levels and vice versa in the 4-h time-scale of the depletion experiments (Fig 1G). Moreover, DIA-induced depletion of cyclin B1 was comparable to the levels of depletion using a pool of four siRNAs 48 h after transfection (Fig 1H). Degron-induced depletion of cyclin B1 was further verified using a separate antibody (Fig 1I).

## Either cyclin B1 or B2 is sufficient to allow proliferation of RPE-1 cells, while cyclin A2 is essential for viability and mitotic entry

We aimed to further reduce the levels of B-type cyclins in these cells and disrupted the CCNB2 gene (encoding for cyclin B2) in B1$^{dd}$ cells by CRISPR editing, producing B1$^{dd}$/B2$^{ko}$ cells. We verified absence of cyclin B2 by immunoblotting (Fig 2A) and genome sequencing of the targeted locus (see Materials and Methods section). To investigate the long-term penetrance of this depletion system, we further analysed A2$^{dd}$, B1$^{dd}$ and B1$^{dd}$/B2$^{ko}$ cells by measuring the distribution of cell cycle phases in asynchronous cells, 24 h after DIA treatment (Fig 2B and C) and long-term proliferation using colony formation assays (Fig 2D). Loss of cyclin A2 caused an accumulation of the 4N population and prevented cell proliferation, but

depletion of cyclin B1 had no discernible effect on the cell cycle phase distribution of RPE-1 cells and only reduced but not prevented colony formation. However, B1$^{dd}$/B2$^{ko}$ cells showed an accumulation of 4N cells and a block in proliferation following DIA treatment (Fig 2B–D) suggesting that these two mitotic cyclins can compensate for each other to support cell viability.

We next measured mitotic entry dynamics in cells following DIA depletion (unless otherwise indicated, DIA treatment for the remaining MS indicates 4-h treatment in asynchronous cells) of either cyclin A2 or both B1 and B2 (Figs 2E and EV1). DIA-treated A2$^{dd}$ cells ceased to enter mitosis rapidly following drug addition, while lack of cyclin B1 and B2 did not measurably affect the dynamics of mitotic entry. However, these cells had severe defects in mitotic progression and cell division (see below). We also tested if cyclin

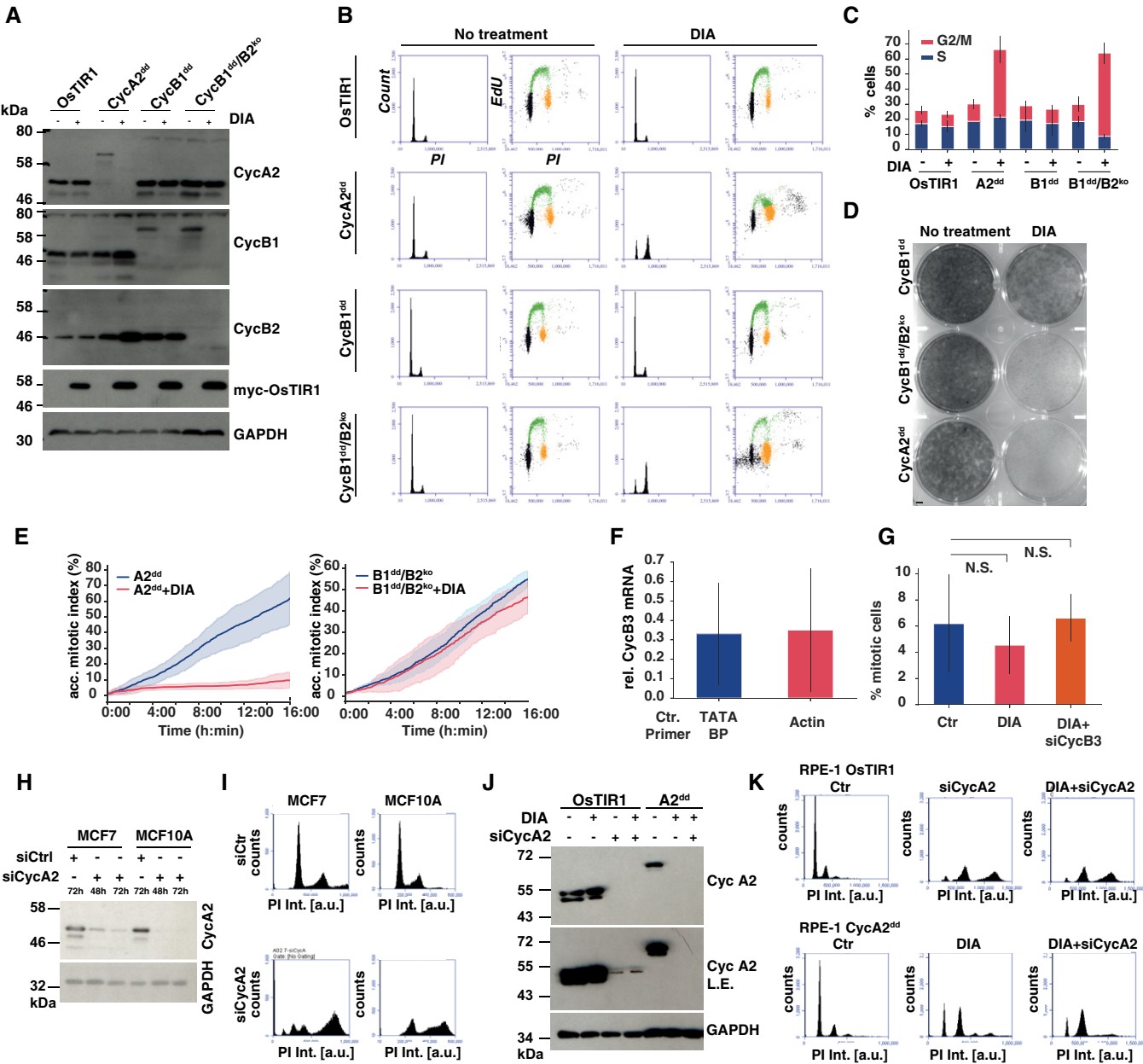

**Figure 2.**

**Figure 2.   Cell proliferation in the absence of cyclin A2 and B-type cyclins.**

A   Cyclin B2 knock-out and induced degradation by immunoblotting. Indicated cell lines were analysed 24 h after mock or Dox/IAA/Asv (DIA) treatment using the indicated antibodies to confirm homozygous gene tagging and efficiency of protein degradation.

B   Cell cycle analysis 24 h after DIA treatment. Cells were analysed by EdU labelling, PI staining and FACS analysis. The histograms show the PI intensities while the dot plots show EdU incorporation (*y*-axis) vs PI intensity (*x*-axis). Gating of cell cycle phases is indicated by colour (G1—black, S phase—green, G2/M phase—orange)

C   Cell cycle phase frequencies quantified from flow cytometry data (shown in (B)) in indicated cell lines 24 h after DIA treatment, (*n* = 3 experiments, s.d. indicated by error bars).

D   Cell proliferation of A2$^{dd}$, CycB1$^{dd}$ and B1$^{dd}$/B2$^{ko}$ following mock or DIA treatment. One thousand cells were plated in each well (diameter 3.5 cm) and incubated for 10 days before methanol fixation and Crystal Violet staining.

E   Kinetics of mitotic entry as measured by time-lapse microscopy in A2$^{dd}$ and B1$^{dd}$/B2$^{ko}$ cells following mock or 4-h DIA treatment of asynchronous cells. The cells were imaged for 16 h with 5-min intervals using widefield DIC; mitotic entry was manually scored by detecting cell rounding. Curves display the cumulative mitotic index (data from three repeats, *n* > 500 cells per condition, s.d. indicated by shaded area).

F   qPCR analysis of cyclin B3 mRNA levels, following 72-h depletion in B1$^{dd}$/B2$^{ko}$ cells. For quantification, we used primers to amplify two control mRNAs, TATA-binding protein and actin. The plot shows the levels of CycB3 siRNA-depleted mRNA relative to Ctr siRNA-transfected cells. (Bars indicate the mean of three independent experiments; error bars indicate the standard deviation between these three repeats.)

G   Mitotic index measurements of B1$^{dd}$/B2$^{ko}$ cells with the indicated treatments. Cells were transfected with siRNA for 72 h; after 36 h, they were blocked for 24 h with Thymidine and fixed 12 h after release from Thymidine. ProTAME and Apcin were added for the final 2 h before fixation. Mitotic cells were scored based on DAPI staining and on condensed chromosome formation. The bar plots show mean values of three biological repeats (*n* = 100 per repeat and sample, error bars indicate standard deviation, and *P*-values were calculated using an independent two-sample t-test).

H   Cyclin A2 siRNA depletion in MCF7 and MCF10A cells. The cells were transfected with Ctr or cyclin A2 siRNA for indicated length of time and probed for cyclin A2 levels by immunoblotting.

I   Cyclin A2 siRNA depletion causes endoreplication. Following 72 h of siRNA transfection, MCF7 and MCF10A cells were analysed by PI staining and FACS. The histograms show the changes in DNA content (PI Int.) towards > 4N following cyclin A2 depletion.

J   Cyclin A2 siRNA and degron depletion in RPE-1 cells. RPE-1 OsTIR1 and RPE-1 A2$^{dd}$ cells were subjected to 72 h of cyclin A2 siRNA depletion and/or of DIA treatment as indicated and probed for cyclin A2 levels by immunoblotting. The longer exposure (L.E.) reveals incomplete depletion of cyclin 2 by siRNA.

K   Cyclin A degron depletion causes accumulation of cells in G2 phase. Following 72 h of siRNA transfection or DIA treatment, RPE-1 OsTIR1 and A2$^{dd}$ cells were analysed by PI staining and FACS. The histograms show the changes in DNA content (PI Int.) towards > 4N following cyclin siRNA A2 depletion in RPE-1 OsTIR1 cells, while DIA treatment in A2$^{dd}$ cells does cause an increase in the 4N but not > 4N peak.

B3 could account for mitotic entry observed in the absence of cyclin B1 and B2. However, we could not detect cyclin B3 protein in these cells by immunoblotting despite the presence of the mRNA of this cyclin. Moreover, siRNA depletion of the cyclin B3 mRNA in combination with DIA treatment of B1$^{dd}$/B2$^{ko}$ cells did not appear to reduce the number of mitotic cells (Fig 2F and G). Taken together, these results suggest that cyclin A2 is essential to initiate mitosis and that B-type cyclins are not rate-limiting for mitotic entry.

Previous experiments suggested that mouse embryonic fibroblasts could proliferate in the absence of cyclin A2, yet it appears to be essential in RPE-1 cells (Kalaszczynska *et al*, 2009; Gopinathan *et al*, 2014). To address these differences, we analysed the effects of cyclin A2 depletion in two breast epithelial cell lines, one cancerous (MCF7) and one un-transformed (MCF10A). Following 72 h of depletion, cyclin A2 levels were substantially reduced (Fig 2H) and these cells appeared to have undergone endoreplication as judged by the increase in the > 4N population (Fig 2I). A similar increase in the > 4N population was observed in RPE-1 OsTIR1 cells following 72 h cyclin A2 depletion, while DIA-mediated degron depletion alone, or in combination with cyclin A2 siRNA caused an increase in the 4N but not > 4N population of RPE-1 cells (Fig 2J and K). These data suggest that cyclin A2 is critical for cell cycle control in several epithelial cell lines. They also demonstrate that due to either slow, or incomplete depletion of cyclin A2 by siRNA, cells undergo endoreplication. In contrast, rapid and acute cyclin A2 depletion results in the accumulation of the 4N population, but does not cause a significant increase in endoreplication.

## Cyclin B is essential for sister-chromatid segregation and cytokinesis

We subsequently analysed the phenotypes of cells lacking B-type cyclins in closer detail. DIA-treated B1$^{dd}$/B2$^{ko}$ cells readily entered mitosis, but failed to undergo sister-chromatid segregation and cytokinesis (Figs 3A and EV2). Indeed, in more than 150 mitotic events observed, we did not record a single successful example of normal mitotic exit in cells lacking cyclin B1 and B2, while cells lacking either B1 or B2 alone did not have apparent mitotic defects. Even in cases where DIA-treated B1$^{dd}$/B2$^{ko}$ cell-initiated segregation and cytokinesis, all cells ultimately failed cell division, resulting in a single bi-nucleated daughter cell. A fraction of these cells did not transit prophase and exited mitosis without undergoing NEBD. Moreover, we observed a delay in mitotic progression following DIA treatment (Fig 3B). To analyse this phenotype with improved resolution, we recorded mitotic progression in B1$^{dd}$/B2$^{ko}$ cells that stably expressed Fusion Red-Histone H2B and the kinetochore component Mis12-GFP. Figure 3C shows a typical example of mitotic progression in cells with or without B-type cyclins (see also Movie EV1 and EV2). While cells lacking cyclin B can clearly initiate chromosome condensation, NEBD and spindle formation, they fail to segregate their sister chromatids and exit mitosis with a single fragmented nucleus.

A simple explanation for this failed mitotic exit could be the early degradation of cyclin A2 by the APC/C that is independent of the SAC. However, we observed a distinct mitotic defect in cells lacking cyclin B1/2 (Fig 3D) that were arrested in mitosis by a APC/C inhibitor cocktail, proTAME/Apcin (Sackton *et al*, 2014). Overall, we observed an enrichment in prophase, and prometaphase (or defective metaphase) after DIA treatment and APC/C inhibition, but failed to identify DIA-treated cells with a clear metaphase alignment. Accordingly, we observed a small but significant decrease in the level of phosphorylation at Ser-Pro residues following DIA treatment as judged by quantitative immunofluorescence (Fig 3E).

ProTAME/Apcin was effective in stabilising cyclin A, as judged by accumulation of this protein in SAC arrested RPE-1 cells

following proTAME/Apcin treatment (Fig 3F). This phenotype was unchanged, even in cells depleted of the essential APC/C subunit Cdc27 and simultaneously treated with APC/C inhibitors, further

suggesting that stabilising cyclin A does not rescue the spindle defects caused by cyclin B depletion (Fig 3G). A typical example of a mitotic spindle in proTAME/Apcin-arrested DIA-depleted cyclin

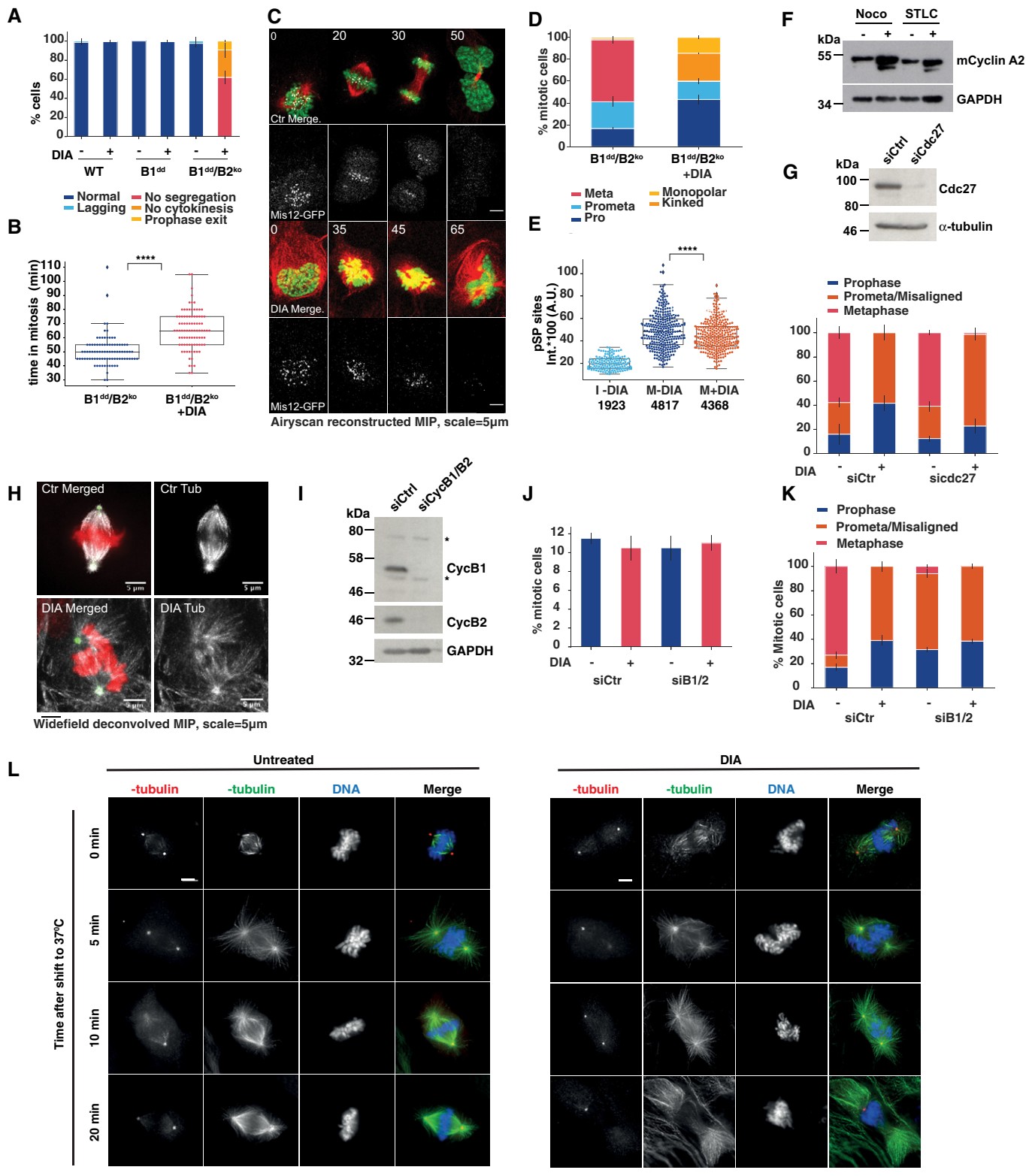

**Figure 3.**

**Figure 3.   Cyclin B is essential for sister-chromatid bi-orientation and segregation.**

A   Frequency of mitotic phenotypes observed in indicated cell lines with or without DIA treatment (three repeats, $n > 50$ per repeat, error bars indicate s.d.). DIA was added for 4 h in asynchronous cells followed by 12 h of imaging.

B   Mitotic duration comparing mock-treated and DIA-treated B1$^{dd}$/B2$^{ko}$ cells. Time of entry and exit was defined by chromosome condensation and decondensation (three repeats, $n > 50$ per repeat, the boxplot indicates median, first and third quartile and minimum/maximum values).

C   Representative images from live-cell imaging of SiR-Tubulin-labelled (red) mitotic B1$^{dd}$/B2$^{ko}$ cells expressing FusionRed-H2B (green), Mis12-GFP (white); time in mins, scale bar = 5 μm. Cells were imaged 4 h after DIA treatment.

D   Characterisation of mitotic spindle in proTAME/Apcin (P/A) arrested B1$^{dd}$/B2$^{ko}$ cells with or without DIA treatment (three repeats, $n > 50$ per repeat, error bars indicate s.d.). For cell synchronisation (P/A synchronisation), we released from 24-h Thymidine block into DIA- or PBS-containing medium and added (P/A) 10 h after release. APC/C inhibition lasted for 2–4 h and was followed by fixation and immunofluorescence analysis.

E   Intensities of anti-(pSP) antibody staining in cells treated as in (D) (three repeats, $n > 50$ per repeat). Key to legend: I-DIA = control interphase cells, M-DIA, M + DIA = control and DIA-treated mitotic cells. The boxplot indicates median, first and third quartile and minimum/maximum values; numbers below panel indicate the median values calculated from each data set.

F   Cyclin A2 levels were assessed by immunoblotting after 18 h in 100 ng/ml nocodazole and/or 5 μM STLC. 6 μM proTAME was added to the indicated samples for the final 6 h.

G   Upper panel, confirmation of Cdc27 depletion by immunoblotting 72 h after siRNA transfection in B1$^{dd}$/B2$^{ko}$ cells. Lower panel, following 40-h siRNA transfection, B1$^{dd}$/B2$^{ko}$ cells were treated for 24 h with Thymidine, released for 10 h and treated with proTAME and Apcin for additional 4 h. At this point, the cells were fixed and stained with tubulin and pericentrin for immunofluorescence analysis. Data are mean values from 3 independent repeats ($n > 50$ for each repeat); error bars show standard deviation.

H   Representative images of mitotic spindles in control- and DIA-treated B1$^{dd}$/B2$^{ko}$ cells following arrest in mitosis by proTAME/Apcin treatment (as in D). Tubulin staining is shown in white, pericentrin in green and DAPI in red, scale bar = 5 μm.

I   Confirmation of cyclin B1 and B2 depletion 48 h after siRNA (pool of four different siRNAs) depletion by immunoblotting in WT RPE-1 cells. Asterisks indicate non-specific bands that are not targeted by siRNAs.

J   Additional siRNA depletion of CycB1 and B2 in Ctr- and DIA-treated B1$^{dd}$/B2$^{ko}$ cells does not prevent mitotic entry. Cells were released from a 24-h Thymidine arrest for 10 h at which point proTAME/Apcin were added for an additional 6 h before fixation, Hoechst staining and mitotic index scoring. Data are mean values from three independent repeats ($n > 100$ for each repeat); error bars show standard deviation.

K   Additional siRNA depletion of CycB1 and B2 in Ctr- and DIA-treated B1$^{dd}$/B2$^{ko}$ cells does not change the mitotic phenotypes caused by DIA depletion. Cells were treated as in (E) and stained with tubulin and pericentrin for immunofluorescence analysis. Data are mean values from three independent repeats ($n > 50$ for each repeat); error bars show standard deviation.

L   Immunofluorescence images from mitotic Ctr- and DIA-treated B1$^{dd}$/B2$^{ko}$ cells after P/A synchronisation stained with anti-alpha-tubulin (green), anti-gamma-tubulin (red) antibodies and DAPI (blue), scale bar = 5 μm. Cells were exposed to ice-cold medium and either fixed immediately after cold exposure, or incubated for 5, 10, or 20 min at 37°C before fixation.

B1$^{dd}$/B2$^{ko}$ cells is shown in Fig 3H. Our observation that cells lacking cyclin B1 and B2 can progress to prometaphase agrees with previous reports based on siRNA depletion experiments (Chen *et al*, 2008; Gong & Ferrell, 2010). To further increase the level of cyclin B depletion and to rule out incomplete gRNA/Cas9-mediated gene depletion of cyclin B2, we combined cyclin B1/2 siRNA- and degron-induced degradation, reasoning that mRNA and protein depletion are expected to synergise. However, even in the case of combined siRNA and degron depletion, we did not see a reduction in mitotic index or changes in the spindle phenotypes (Fig 3I–K). This further supports the notion that substantial depletion of B-type cyclins does not affect mitotic entry dynamics, but prevents the establishment of metaphase alignment and sister-chromatid segregation.

Exposure to ice-cold medium revealed intact K-fibres in DIA-treated B1$^{dd}$/B2$^{ko}$ cells, and re-exposure to warm medium resulted in the reformation of a disorganised spindle with extended astral microtubules (Fig 3L). We also used the centromere/kinetochore markers CenpA and CenpB to assess bi-orientation and segregation of sister chromatids (Fig EV3). The markedly reduced distance between sister centromeres following DIA treatment suggests lack of tension. Moreover, we frequently observed unseparated centromere pairs in cells that were exiting mitosis, as judged by attempted furrow formation, indicating that separase activation and/or microtubule force generation was impaired in these cells. These distinct phenotypes suggest that while cyclin B is required for specific regulatory events to coordinate the metaphase/anaphase transition, it is not essential for the establishment of the mitotic state.

**Cyclin A2 is required in G2 phase to trigger mitotic entry**

Previous reports implicated cyclin A2 in triggering M phase via Cdk1 activation (Fung *et al*, 2007; Gong & Ferrell, 2010), but could not differentiate between knock-on effects of DNA replication problems and G2-specific functions of cyclin A2. We took advantage of the A2$^{dd}$ cells to address this question by performing G2-specific depletion of cyclin A2. For this purpose, we used a PCNA-mRuby fusion (Zerjatke *et al*, 2017). PCNA forms foci in S phase but these replication structures dissipate upon completion of DNA replication (Leonhardt *et al*, 2000) (Fig 4A). This allowed us to identify cells that were DIA-treated after completion of S phase. This effectively blocked further progression into mitosis (Fig 4B), suggesting that either the presence, or new synthesis of cyclin A in this cell cycle phase is required for mitotic entry. To further support this notion, we performed EdU pulse labelling at the time of DIA treatment, followed by fixation and CENP-F immunofluorescence 4 h later to mark cells that were in G2 phase (CENP-F-positive/EdU-negative) when induced cyclin A2 degradation began (Fig 4C). This resulted in an accumulation of cells that had been in G2 phase at the time of DIA treatment, further supporting a G2-specific role for cyclin A2. Inhibition of Wee1 with the small molecule inhibitors MK1775 (Hirai *et al*, 2009) or PD-166285 (Wang *et al*, 2001) was sufficient to overcome this block and resulted in resumption of mitotic entry with kinetics comparable to the controls (Fig 4D–F). Once the G2 block was overcome by Wee1 inhibition, cells lacking cyclin A could form a metaphase spindle and undergo chromosome segregation and cell division after delaying mitotic progression in metaphase for ~ 20 min (Fig 4F). Likewise, siRNA depletion of B55 alpha and delta

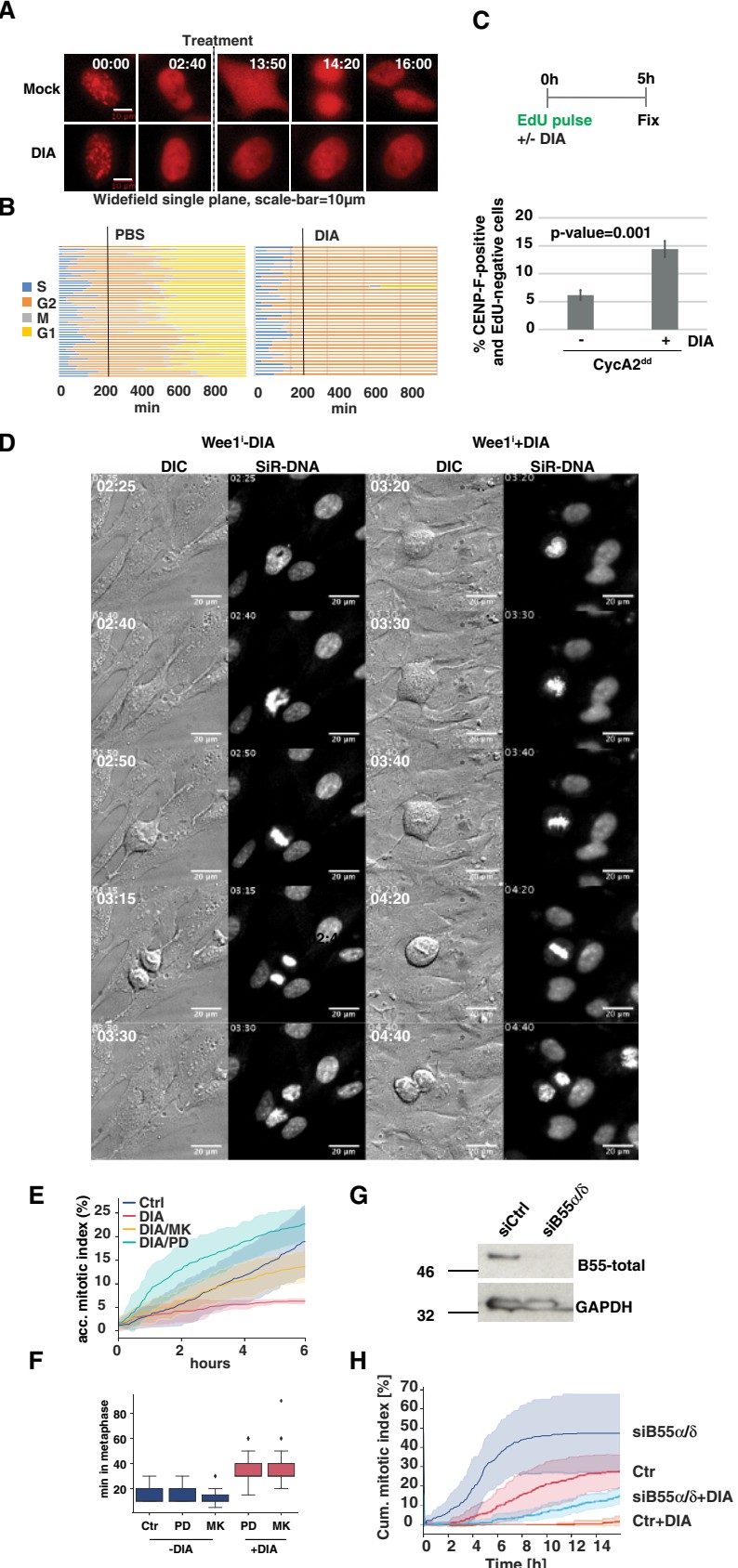

Figure 4.

**Figure 4. Cyclin A2 is essential to trigger mitotic entry in G2 phase.**

A  Time-lapse images of PCNA-mRuby-tagged A2[dd] cells. The imaging sequence was started at the time of doxycycline addition, and degron activity was triggered 3 h later by addition of IAA and Asv or PBS (indicated by dashed line). Time is shown as hh:min; scale bar equals 10 μm. For further analysis, cells were chosen that had dissipated their PCNA foci before the addition of DIA to ensure that cyclin A2 degradation was triggered in G2 phase.

B  Single-cell analysis of 40 cells per condition treated as described in (A); each line represents the timing of G2/M/G1 progression for a single cell.

C  A2[dd] cells were pulsed with EdU to label S phase then protein degradation was induced by DIA addition for 5 h and before fixation and Edu/CENPF/DAPI staining. Cells that were in G2 at the time of DIA or mock treatment were identified as CENPF-positive, EdU-negative. Mitotic cells were excluded based on nuclei morphology. The bar plots show mean percentage of EdU-negative/CENPF-positive cells of three repeats; error bars indicate standard deviation.

D  Images from time-lapse sequence of mitosis showing cell division in controls and after addition of 0.5 μM PD-166285 in DIA-treated A2[dd] labelled with SiR-DNA (time is indicated as hh:min; the scale bars represent 20 μm).

E  Accumulative index of mitotic entry measured by time-lapse microscopy in A2[dd] cells following mock or DIA treatment and addition of Wee1 inhibitors MK1775 (1 μM) or PD-166285 (0.5 μM). Asynchronous cells were DIA-treated for 4 h and then imaged for 6 h in 5-min intervals. (Data from three repeats, $n > 500$ cells per condition; standard deviation is indicated by shaded area).

F  Quantification of metaphase duration in A2[dd] cells treated with DIA and Wee1 inhibitors as indicated ($n > 50$ per repeat, the boxplot indicates median, first and third quartile and minimum/maximum values.)

G  A2[dd] cells were transfected with Ctr or B55α/δ siRNA. Forty-eight hours after transfection, the cells were treated with Thymidine for 24 h and then released. Ten hours later, the cells were probed for B55 depletion by immunoblotting using a pan-B55 monoclonal antibody.

H  Lack of B55 partially rescues mitotic entry in DIA-treated A2[dd] cells. Cells transfected with Ctr siRNA or B55α/δ siRNA as in (B) were released from Thymidine, mock- or DIA-treated and analysed by widefield live-cell imaging 8 h after the release. The cells were imaged every 10 min for 15 h and manually scored for mitotic index. The lines represent mean values of three independent experiments, and standard deviation is indicated by the coloured areas ($n > 100$ per experiment).

resulted in a partial rescue of mitotic entry in DIA-treated A2[dd] cells following release from a Thymidine arrest (Fig 4G and H).

### Nuclear cyclin B1 partially compensates for S and G2 defects in cells lacking cyclin A2

We went on to analyse the effects of acute cyclin A depletion on cell cycle progression using an Edu pulse-chase experiment. We labelled asynchronous A2[dd] cells with EdU after 4-h DIA treatment, removed the EdU from the medium and followed these cells for 12 additional hours (Fig 5A–C). This analysis revealed that DIA-treated A2[dd] cells can progress from G1 to S phase (cells accumulating in Gate 1), but do not progress from G2 phase to the next G1 phase (cells leaving Gate 2). Moreover, we observed a significant delay in the progression from early S phase (Gate 3) to mid (Gate 4)- and late (Gate 5) S phase. Cyclin A2-depleted cells that were in S phase at the time of EdU labelling did not undergo cell division to reach the next G1 phase in the subsequent 12 h (re-accumulation of cells in Gate 3). Taken together, these data suggest that cyclin A2 is critical both for the G2/M transition and also to complete DNA replication, but not essential for progression from G1 to S phase.

We next addressed, if an increased expression of cyclin B1 can compensate for the loss of cyclin A. We generated constructs to express YFP, cyclin B1-YFP (B1-WT), and also targeted cyclin B1 to the nucleus by fusing it with a SV40 nuclear localisation sequence (Collas & Aleström, 1996) (B1-NLS). We used the sleeping beauty transposon system (Kowarz *et al*, 2015) to engineer existing A2[dd] and B1[dd]/B2[ko] cell lines by incorporating the constructs with a doxycycline-inducible promoter into the genome. This allowed us to rapidly deplete the endogenous cyclin A2 or B1 and compensate this with newly expressed cyclin B1 tagged with either YFP or YFP-NLS (Figs 5D and EV4). Both WT and nuclear cyclin B1 showed the expected cytoplasmic and nuclear localisation and fully reverted the sister-chromatid segregation and cytokinesis defects in DIA-treated B1[dd]/B2[ko] cells (Fig EV4). This suggested that both incorporated constructs are generating functional cyclin B1, whereas the YFP control showed no difference in phenotype, as expected. We then analysed the effects of induced cyclin B1-YFP expression in DIA-treated A2[dd] cells using the same EdU pulse-chase assay described

above (Fig 5E and H). Induced expression of YFP alone did not change the cyclin A2 depletion phenotypes and led to a delay in S-phase progression and a block in the appearance of new G1 cells, while in WT cells over 80% of Edu-positive cells progressed into the next G1 phase. Both WT and nuclear cyclin B1-YFP could induce a reversion of this phenotype with about 10% of cells reaccumulating in the next G1 phase with WT cyclin B1-YFP and 30% with nuclear cyclin B1-YFP. We then analysed progression of these cells following release from a Thymidine arrest by live-cell imaging (Fig 5F). Similar to the pulse-chase analysis, nuclear cyclin B1-YFP was approximately threefold more successful than WT cyclin B1-YFP in replacing cyclin A2. Cells expressing nuclear cyclin B1-YFP could promote cell division at a rate of ~ 50% lower compared to cells expressing endogenous cyclin A2. Similar to the A2[dd] cells treated with Wee1 inhibitors, we observed a significant delay in metaphase when mitosis was induced by over-expressed cyclin B1 in cyclin A2-depleted cells (Fig 5G).

These data suggest that cyclin A2 is required for both completion of DNA replication and initiation of mitosis. These defects can be partially compensated by nuclear cyclin B1, suggesting that the nuclear localisation is a key factor in explaining the essential functions of cyclin A2, as previously suggested by siRNA depletion experiments (Gong *et al*, 2007). Cells entering mitosis in the absence of cyclin A2 are partially defective in mitotic progression suggesting additional functions of this cyclin in orchestrating sister-chromatid segregation, as previously suggested (Kabeche & Compton, 2013; Dumitru *et al*, 2017). One possible consequence of cyclin A2 depletion could be that free Cdk2 sequesters cyclin B thereby generating a dominant negative effect on mitotic entry. This is unlikely to be the case for the observed G2 arrest, because cyclin B1-YFP immunoprecipitates did not show an increase in co-precipitated Cdk2 (Fig 5I).

### Cyclin B is required for the phosphorylation of a specific subset of proteins in mitosis

Given the failure of cells lacking cyclin B to align the metaphase spindle and to initiate sister-chromatid segregation, we aimed to further quantify the contribution of cyclin B to mitotic phosphorylation events. To this end, we identified cyclin B-specific substrates

by comparing the phospho-proteome in mitotic control and DIA-treated B1$^{dd}$/B2$^{ko}$ cells (Fig 6A–C). We quantified differences in phosphorylation in 5,670 phosphorylation sites (Table EV1), including 2,351 that have a [S/T]-P motif in the peptide sequence detected, and identified 197 sites in 137 substrates that were

reduced by ~ 2-fold or more. Thus, cyclin B depletion causes a significant but relatively minor effect on the majority of mitotic phosphorylations. We observed a more dramatic reduction in phosphorylation of only ~ 3.5% of the phospho-proteome and ~ 5.6% of the proline-directed sites (131/2,351). These substrates were

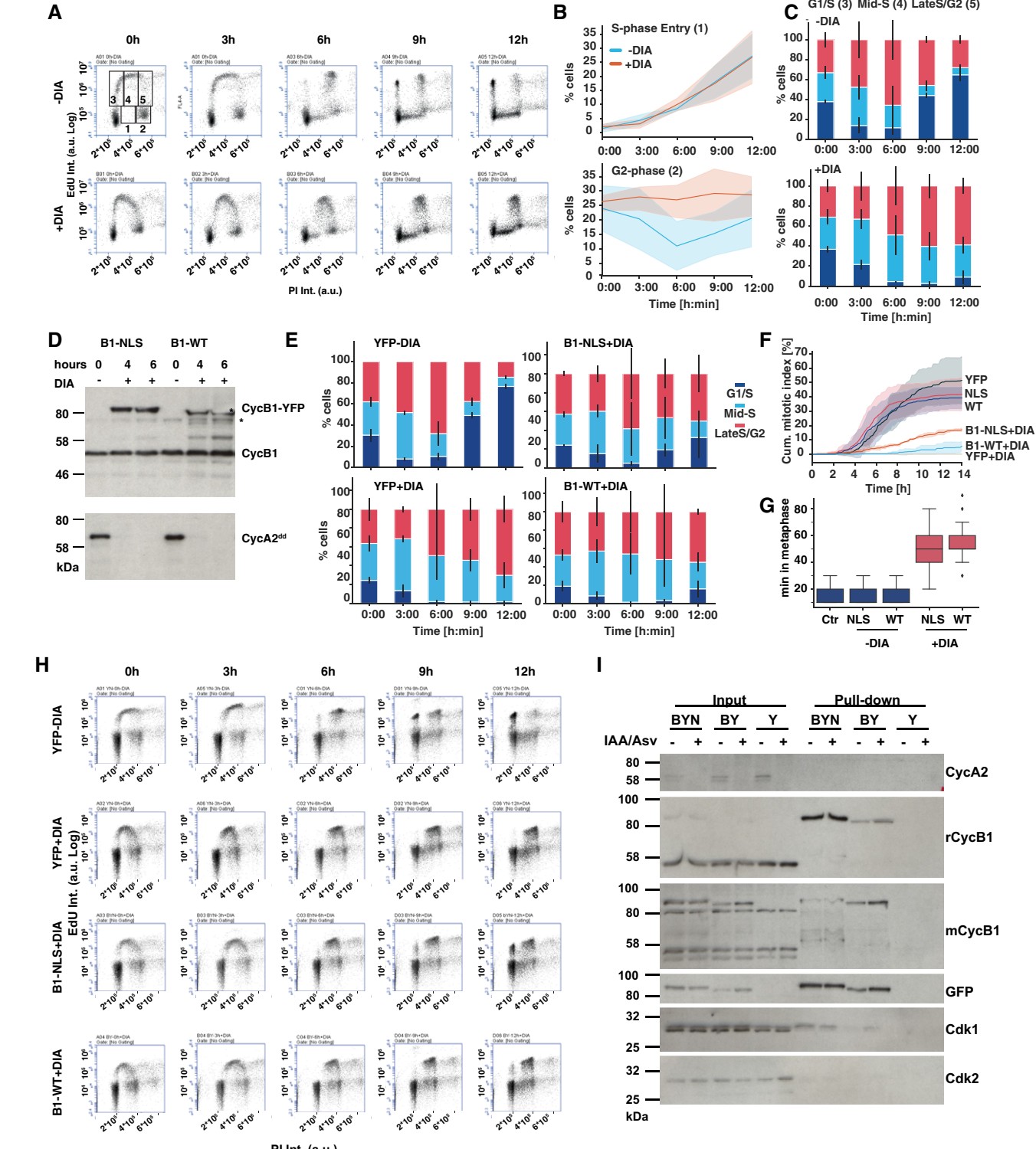

**Figure 5.**

**Figure 5.   Nuclear cyclin B1 partially compensates for cyclin A2.**

A   FACS plots from EdU pulse-chase experiment. Cells were DIA- or mock-treated for 4 h and then labelled with EdU for 1 h. The EdU was then washed out and samples were taken every 3 h for 12 h and analysed by EdU/PI staining and FACS. The top left panel shows the gates that were used for analysis. The quantifications shown in the panels to the right are from three independent repeats of this experiment.

B   Quantification of cells progressing into Gate 1 (top panel, new S-phase cells) and Gate 2 (bottom panel, G2 cells) over time. The lines represent the mean of three experiments; the coloured areas indicate the standard deviation.

C   Relative changes in the percentage of EdU-positive cells in gates 3, 4 and 5 (G1/early S, mid-S and late S/G2 phase). The bars indicate the relative mean values from three independent experiments; the error bars represent the standard deviation.

D   Cyclin A2$^{dd}$ cells with inducible CycB1-YFP (B1-WT) and CycB1-YFP-NLS (B1-NLS) were analysed for DIA-induced cyclin B1 expression/cyclin A2 depletion. Samples were collected at indicated time points and probed by immunoblotting with cyclin B1 and cyclin A2 antibodies (* indicates non-specific band).

E   Quantifications of EdU pulse-chase data as in (C) in DIA-treated cyclin A2 cells expressing B1-WT, B1-NLS or YFP as indicated. The bars indicate the relative mean values from three independent experiments; the error bars represent the standard deviation.

F   Accumulative mitotic index measurement of cyclin A2 cells following cyclin A2 depletions and induction of indicated cDNAs. The lines represent the mean of three experiments; the coloured areas indicate the standard deviation.

G   Quantification of metaphase duration in A2$^{dd}$ cells treated with DIA following induction of indicated proteins ($n > 50$ per repeat, the boxplot indicates median, first and third quartile and minimum/maximum values).

H   FACS plots from EdU pulse-chase experiment in A2$^{dd}$ cells following induction of indicated genes. Data are representative example for quantifications shown in (E).

I   Immunoprecipitation of YFP from extract from A2$^{dd}$ following induction of YFP (Y), CycB1-YFP (BY) and CycB1-YFP-NLS (BYN). The samples in the odd lanes (IAA/Asv−) were treated with 1 µg/ml doxycycline alone, while samples in even lanes (IAA/Asv+) were treated with Auxin and Asv to induce cyclin A2 depletion. Total cell extract and immune precipitates were probed by immunoblotting with the indicated antisera.

predominantly cytoskeletal and chromatin-associated proteins (Fig. 6D). A more detailed network analysis (Fig EV5) identified a variety of mitosis-associated processes that involve cyclin B-specific substrates, including chromosome structure, microtubule (MT) dynamics and cytokinesis.

To test if this set of cyclin B-specific phosphorylation sites correlates with specific mitotic functions for cyclin B, we analysed the localisation of a group of cyclin B-specific targets identified in the screen. For this purpose, we chose Ki-67, the chromosome passenger complex (CPC) members INCENP and Borealin, CDCA2, Top2B and Tpx2. Each of these proteins shows a significant loss of phosphorylation after DIA treatment in the mass spectrometry analysis (Fig 6E, Table EV2). Ki-67 acts as a surfactant on condensed mitotic chromosomes (Cuylen *et al*, 2016) and is heavily phosphorylated by Cdk1 in mitosis (Blethrow *et al*, 2018). Loss of cyclin B following DIA treatment resulted in a loss of Ki-67 from the chromosome periphery

**Figure 6.   Mitotic cyclin B-dependent phosphorylation sites.**

A   Cell synchronisation scheme: to obtain an optimal enrichment in mitotic cells, we pre-synchronised B1$^{dd}$/B2$^{ko}$ cells by serum starvation for 48 h followed by a release into Thymidine for 24 h. Cells were then released into S phase in the presence or absence of DIA and blocked in mitosis by the addition of proTAME. Mitotic cells were collected by shake-off 12 h following the release.

B   Phospho-proteomics workflow. Protein extracts were prepared from the indicated cells, digested with trypsin and labelled with 6-plex TMT reagents. The TMT-labelled peptides were then mixed and subjected to fractionation by high pH reverse phase into 24 fractions. From each fraction, 95% was taken for phospho-enrichment and 5% retained for analysis of unmodified peptides. For phospho-enrichment, the 24 fractions were combined into a total of 16 fractions. Samples (unmodified and phospho-enriched) were then analysed by LC-MS/MS on an Orbitrap Fusion instrument using synchronous precursor selection (SPS). This analysis resulted in detection of 11,234 phosphorylation sites, of which 5,670 were quantitated. The number of class I sites are indicated in brackets (site localisation probability > 0.75).

C   Quantification of cyclin B-specific substrates by phospho-proteomics. Phosphorylation sites deemed significantly changing are shown in red (*P*: 0.05, 1.96*standard deviation fold change cut-offs, see Material and Methods).

D   Gene ontology analysis of cyclin B-dependent substrates.

E   Quantification of changes in selected cyclin B-dependent mitotic phosphorylations including Ki-67 (left panel), INCENP/Borealin (middle panel) and Top2B, CDCA2 and Tpx2.

F   Representative images of Ki-67 immunofluorescent staining of chromosome spreads from control (Ctr) and DIA-treated P/A synchronised B1$^{dd}$/B2$^{ko}$ cells and quantification of Ki-67 staining intensity using cross sections of chromosomes (data from three repeats, *n* = 50, shaded area indicates standard deviation, scale bar = 5 µm).

G   Representative images from live-cell imaging of DIA-treated SiR-Tubulin-labelled (red) B1$^{dd}$/B2$^{ko}$ cells expressing FusionRed-H2B (green) and AurB-GFP (white); time is indicated in minutes, scale bar = 5 µm. Bottom panel shows frequencies of aberrant AurB localisation in P/A synchronised B1$^{dd}$/B2$^{ko}$ cells. (Bars indicate mean of three repeats, *n* = 50 cells per repeat; stdv is indicated by error bars.)

H   CDCA2 is excluded from mitotic chromosomes in control cells while this exclusion is lost in DIA-treated B1$^{dd}$/B2$^{ko}$ cells. The upper panel shows immunofluorescence images; for quantification, we analysed fluorescence intensity across chromosomes. The data show the mean of 150 cells (from three repeats *n* = 50) and error bars indicate standard deviation.

I   Top2B levels in mitotic cells are increased in DIA-treated B1$^{dd}$/B2$^{ko}$ cells. The left panel shows immunofluorescence images; for quantification, we segmented mitotic DAPI staining, measured the mean intensity of Top2B staining and plotted the ratio between mitotic and interphase nuclei. (Bars indicate median, first and third quartile; data are from three repeats, *n* = 50 cells per repeat.)

J   Tpx2 shows reduced spread on mitotic chromosomes in DIA-treated B1$^{dd}$/B2$^{ko}$ cells. The top panel shows immunofluorescence images; for quantification, we measured the mean intensity of Tpx2 staining on the centrosomes and spindle and plotted the ratio of these values per cell. (Bars indicate median, first and third quartile; data are from three repeats, *n* = 50 cells per repeat.) We also analysed the centrosome/spindle distribution of AurA that does not change following DIA treatment.

Data information: For images in (H–J) the scale bar equals 10 µm.

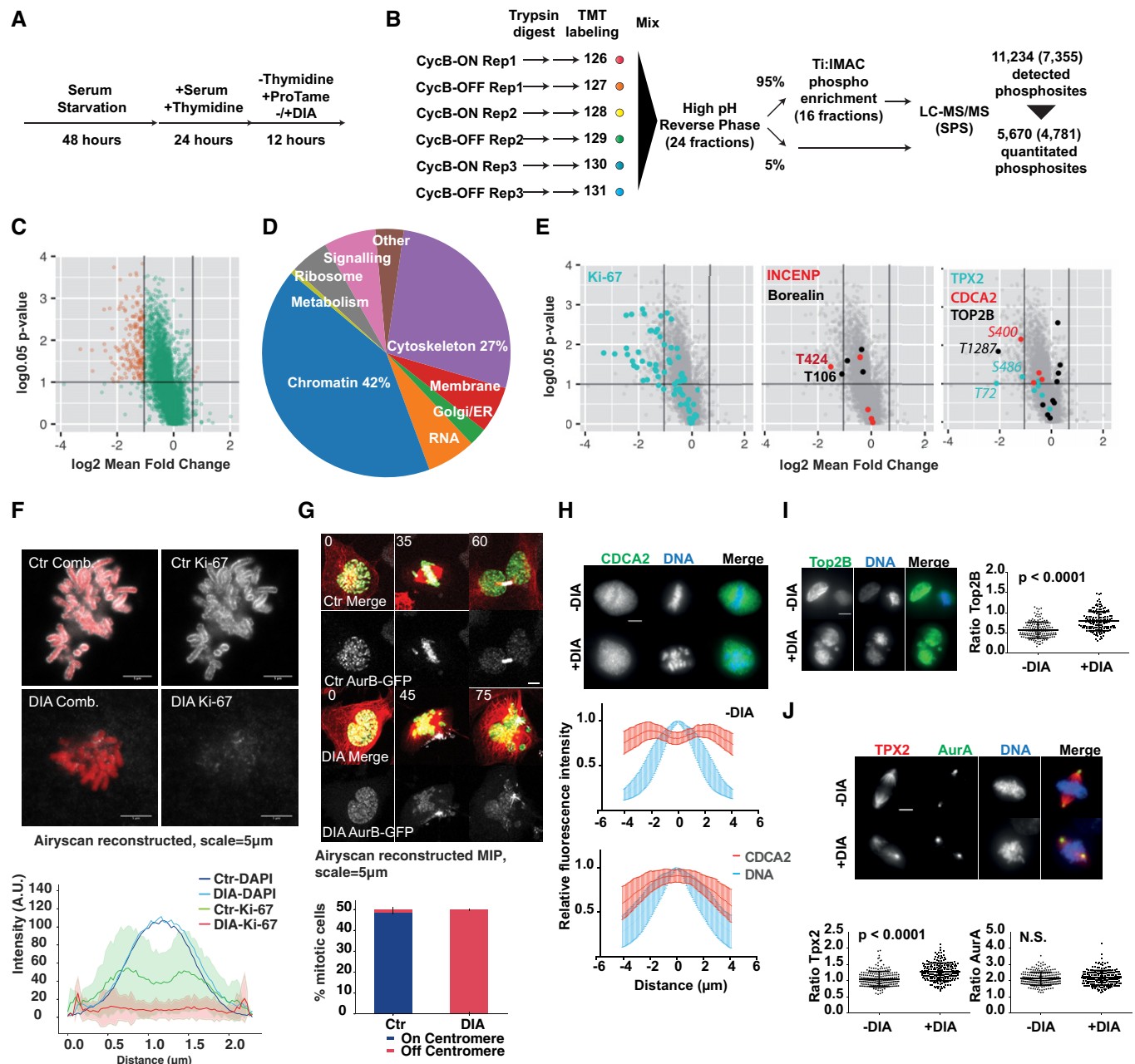

(Fig 6F), correlating with the marked decrease in multiple phosphorylations in Ki-67 in DIA-treated B1$^{dd}$/B2$^{ko}$ cells. Likewise, INCENP and Borealin have previously been identified as mitotic targets of Cdk1 (Goto *et al*, 2005; Daub *et al*, 2008; Dephoure *et al*, 2008; Tsukahara *et al*, 2010), but the cyclin B-specific phospho-sites that we have identified here (T424 in INCENP and T106 in Borealin) have, to our knowledge, not previously been related to CPC localisation and function. DIA treatment resulted in premature displacement of a key CPC subunit, Aurora B-GFP, from the centromeres (Fig 6G, Movie EV3 and EV4) in DIA-treated B1$^{dd}$/B2$^{ko}$ cells, even when cells were arrested in mitosis by APC/C inhibition (Fig 6G bottom panel). CPC mis-regulation also correlates with the significant decrease in histone H3 S10 phosphorylation, a target of AuroraB kinase activity

that we observed (Table EV2). Likewise, the reduction of phosphorylation of CDCA2, Top2B and Tpx2 following DIA treatment correlated with changes in mitotic localisation (Fig 6H–J), further validating the significance of our proteomic data set.

## Regulation of mitotic PP1 and PP2A:B55 activity occurs independently of cyclin B

Our proteomic analysis suggests that cells robustly buffer against loss of Cdk1:cyclin B and allow the phosphorylation of roughly 90% of mitotic Cdk1 substrates in the presence of a single mitotic cyclin, namely cyclin A2. Given that Cdk1 is essential to trigger mitotic phosphorylation and cyclin A2 is the major Cdk1 regulator that can

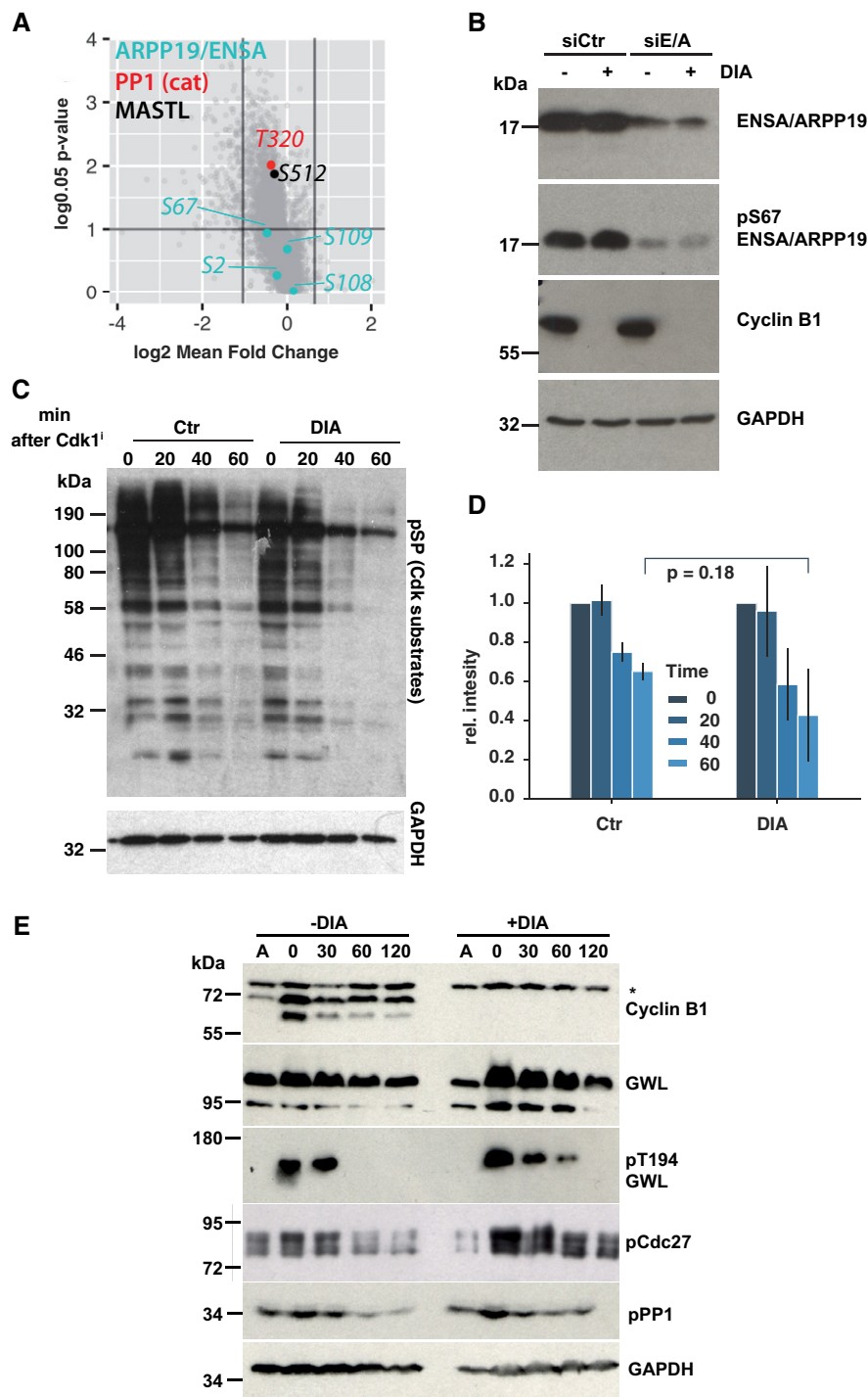

**Figure 7. Mitotic Greatwall, ENSA and PP1 regulation are independent of cyclin B.**

A Quantification of Greatwall, ENSA/ARPP19 and PP1 phosphorylation based on phospho-proteomic data.

B Confirmation of ENSA/ARPP19 S67 phosphorylation in mitotic extracts from Ctr- and DIA-treated cells, before and after ENSA/ARPP19 siRNA depletion.

C Dynamics of mitotic substrate dephosphorylation triggered by Cdk1 inhibition in control and DIA-treated B1$^{dd}$/B2$^{ko}$ cells. The cells were enriched in mitosis by Thymidine release and proTAME/Apcin treatment and treated with 1 μM flavopiridol to trigger mitotic exit. Dephosphorylation was measured by probing the immunoblots with anti Cdk1 substrate (anti-pSP) antibody.

D Three repeats of the experiment in (C) were quantified and corrected based on the corresponding GAPDH intensity values. The bar plots show the mean of three experiments, and the standard deviation is indicated by error bars.

E Dynamics of Greatwall, PP1 and Cdc27 dephosphorylation following Cdk1 inhibition in proTAME/Apcin-treated mitotic cell extracts from Ctr- or DIA-treated cells. Cell extracts were prepared at the indicated times following treatment with 1 μM flavopiridol and analysed by immunoblotting (* indicates non-specific band).

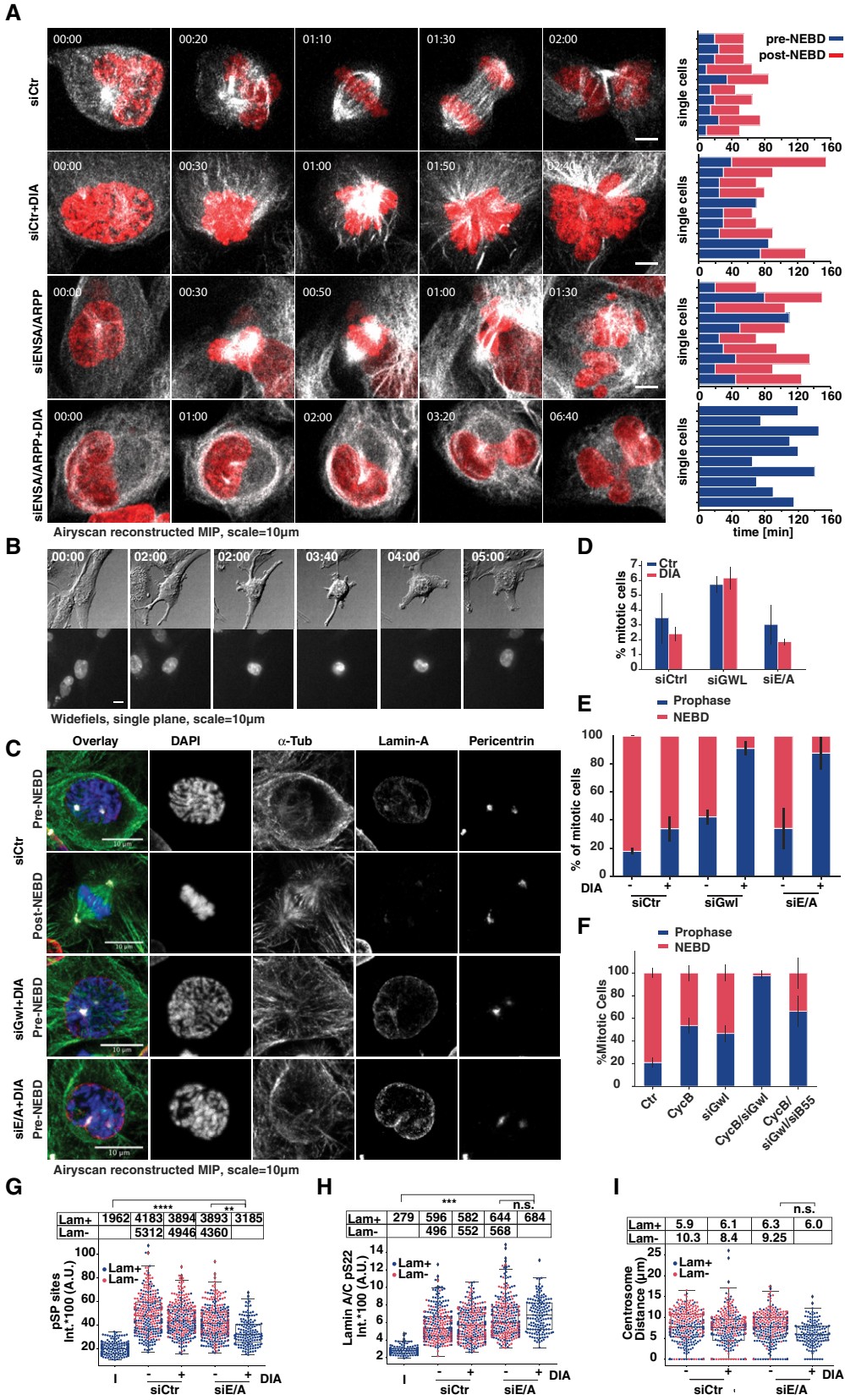

**Figure 8.**

**Figure 8. Greatwall and cyclin B synergise in triggering NEBD but not prophase.**

A  Representative images from live-cell imaging of siRNA-transfected B1$^{dd}$/B2$^{ko}$ cells (FusionRed-H2B in red, SiR-Tubulin in white, scale bar = 10 μm). Bar plot panels on the right show single-cell analysis of 10 cells manually scored for length mitosis pre-NEBD and post-NEBD. Entry into prophase was scored by cell rounding and NEBD was identified by influx of Tubulin in the nucleus.

B  Widefield imaging. DIC (Grey), SiR-DNA (b/w), of ENSA/ARPP19 siRNA-transfected DIA-treated B1$^{dd}$/B2$^{ko}$ cells. Time is indicated in hh:min, scale bar = 10 μm.

C  Panels of representative immunofluorescence images of siRNA-transfected and P/A-treated cells treated as indicated, scale bar = 10 μm.

D  Quantification of mitotic index of siRNA-transfected P/A synchronised cells (See Fig. 4B for representative images; mean value of three repeats, n > 500 per repeat; error bars indicate standard deviation).

E  Quantification of mitotic B1$^{dd}$/B2$^{ko}$ cells following siRNA transfection to deplete Greatwall (siGwl) or ENSA/ARPP19 (siE/A) and/or DIA treatment. Frequencies of mitotic cells with intact (blue) and disassembled (red) Lamin A/C staining are plotted (mean value of three repeats, n > 50 per repeat; error bars indicate standard deviation).

F  As in (E) quantification of control- and DIA-treated mitotic B1$^{dd}$/B2$^{ko}$ cells following siRNA transfection to deplete Greatwall and B55α/δ. Frequencies of mitotic cells with intact (blue) and disassembled (red) Lamin A/C staining are plotted (mean value of three repeats, n > 50 per repeat; error bars indicate standard deviation).

G  Intensity of pSP-antibody staining in siRNA-transfected and DIA-treated B1$^{dd}$/B2$^{ko}$ cells. The swarm-blots classify data from mitotic cells with intact (blue) or disassembled (red) Lamin A/C.

H  As in (G) intensity of anti-Lamin A/C pS22 antibody staining in siRNA-transfected and DIA-treated B1$^{dd}$/B2$^{ko}$ cells.

I  Same as in (G) and (H), data for centrosome distance in mitotic cells. Data for G-I are from three repeats, n > 50 per repeat; the bars indicate median, first and third quartile as well as minimum and maximum values; data are from three repeats, n = 50 cells per repeat).

compensate for cyclin B in late G2 phase, the observed cyclin B-independent phosphorylation events are likely exerted by Cdk1:cyclin A2. The question remains: How cells can cope with loss of the major mitotic kinase Cdk1:cyclin B and still readily phosphorylate most of the mitotic phospho-proteome, including many bona-fide Cdk sites? We hypothesised that this is due to the inactivation of antagonising phosphatase PP2A:B55 in the absence of Cdk1:cyclin B activation. This would suggest that there are two independent mechanisms to promote mitotic Cdk1 phosphorylation: (i) by activating Cdk1:cyclin B, (ii) by activating Greatwall that will phosphorylate ENSA/ARPP19 to inhibit PP2A:B55. In this case, Greatwall phosphorylation as well as the downstream phosphorylation of ENSA/ARPP19 at Ser67 should be fully established in cells lacking B-type cyclins. This notion is indeed supported by our phospho-proteomic data (Fig 7A). We also readily detected ENSA/ARPP19 Ser67 phosphorylation in mitotic extracts of DIA-treated B1$^{dd}$/B2$^{ko}$ cells (Fig 7B). Moreover, dephosphorylation dynamics of various Cdk1 sites and overall Ser-Pro phosphorylations are not significantly changed in proTAME/Apcin-arrested mitotic B1$^{dd}$/B2$^{ko}$ cells following Cdk inhibition, when Ctr- and DIA-treated cells are compared (Fig 7C–E).

**Greatwall and cyclin B synergise to trigger NEBD**

If inhibition of PP2A:B55 is responsible for supporting mitotic phosphorylation in the absence of cyclin B, simultaneous depletion of B-type cyclins and the Greatwall pathway should ultimately prevent cells from entering mitosis. To test this hypothesis, we depleted ENSA and ARPP19 from B1$^{dd}$/B2$^{ko}$ cells and monitored mitotic progression by time-lapse microscopy (Fig 8A, Movie EV5 and EV6). Depletion of ENSA/ARPP19 did not prevent mitotic entry, but resulted in sister-chromatid misalignment and cytokinesis defects, as previously reported for cells lacking Greatwall. Cyclin B depletion by DIA treatment caused similar mitotic exit phenotypes as described above (Fig 3A–D). Surprisingly, cells with simultaneous depletion of ENSA/ARPP19 and cyclin B appeared able to enter prophase, as judged by cell rounding, centrosome separation and compaction of the nucleus. However, these cells did not undergo NEBD and often remained in this prophase-like state for a prolonged period of time, before reverting to interphase (Fig 8A, bottom panel, Fig 8B). This release from the prophase state could be either due to

APC/C activation or slow dephosphorylation of the prophase substrates.

These results suggest that cells can enter prophase in the absence of cyclin B and with high PP2A:B55 activity, but fail to move past NEBD. To further substantiate this observation, we analysed the mitotic state following ENSA/ARPP19 or Greatwall depletion in mitotic B1$^{dd}$/B2$^{ko}$ cells by immunofluorescence (see representative images in Fig 8C). The mitotic index, as judged by cell rounding, centrosome separation and chromosome condensation, was only mildly reduced in the co-depleted cells (Fig 8D). However, more than 90% of mitotic cells lacking both cyclin B and Greatwall or ENSA/ARPP19 failed to undergo NEBD as judged by Lamin A/C staining (Fig 8E). This reduction in cells undergoing NEBD was due to PP2A:B55 activity, because co-depletion of Greatwall and B55α/δ resulted again in an increase in cells that reached the prometaphase state (Fig 8F).

Quantification of various mitotic indicators further supported the observation that cells with high PP2A:B55 and low Cdk1:cyclin B activity reach prophase. Thus, levels of serine–proline phosphorylation were clearly increased compared to interphase cells and only marginally decreased compared to control prophase cells (Fig 8G). Phosphorylation of Lamin A/C S22, a residue that is phosphorylated in late G2/early prophase (Ly et al, 2017), was detected at a level comparable to control prophase and mitotic cells (Fig 8H). Likewise, centrosomes were at a stage of separation similar to controls before NEBD (Fig 8I), suggesting that these cells indeed remained in a prophase-like state without progressing past NEBD.

# Discussion

Our results are summarised by a model (Fig 9A) that we tested by numerical simulation (Fig 9B and C). We propose a simple ordering mechanism for mitotic entry, based on the increasing requirement for Cdk1:cyclin activity, and increasing levels of sensitivity to the antagonising phosphatase activity. This fits well with a similar biochemical model for the ordering of mitotic exit (Bouchoux & Uhlmann, 2011; Cundell et al, 2016). Cyclin A2 in G2 phase is critical to trigger mitosis by counteracting Wee1 and PP2A:B55, because either Wee1 inhibition or B55 depletion can overcome the G2 arrest caused by cyclin A2 depletion. In the absence of Greatwall and

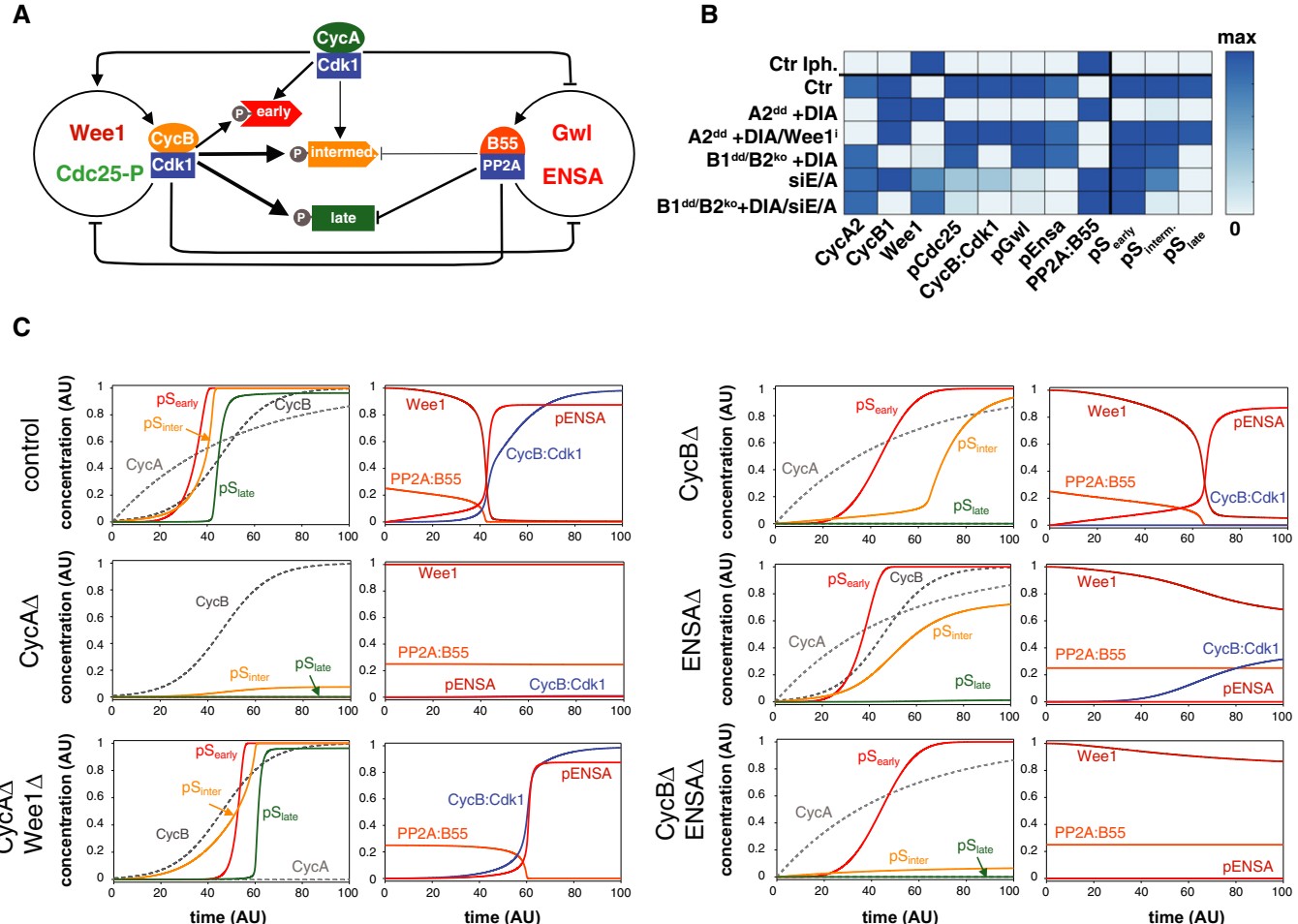

**Figure 9. Numerical Model for cyclin A and B specific and overlapping functions during mitotic entry and progression.**

A Influence diagram of the network controlling the phosphorylation of mitotic substrates. The auto-activation of CycB:Cdk1 and PP2A:B55 is controlled by inhibitory phosphorylation and by the Greatwall-ENSA pathway, respectively. CycA:Cdk1 helps CycB:Cdk1 auto-activation and inhibits PP2B:B55 activity through Greatwall phosphorylation. Three different classes of mitotic substrates (early, intermediate and late) are distinguished based on their sensitivity to CycA:Cdk1, CycB:Cdk1 and PP2A:B55.

B Final level of mitotic regulators and Cdk1-substrates in response to cyclin synthesis. The heat map is based on model simulations (C) starting with initial conditions without cyclins (interphase, top row) until high and constant cyclin levels caused by APC/C inhibition.

C Numerical simulation of mitotic substrate phosphorylation network in response to rising cyclin levels. The model simulates mitotic entry experiments where cyclin levels become stabilised by APC/C inhibition. Each row corresponds to one of the experimental conditions used in this work (control, CycA depletion etc.). Left columns: cyclin levels and phosphorylated mitotic substrates (early, intermediate and late). Right columns: temporal changes of some mitotic regulators. Changes of Cdc25-P and Gwl-P (not shown) are similar to phosphorylated ENSA (pENSA).

cyclin B, cells enter a prophase state for several hours before reverting back to interphase, due to either APC/C activation or dephosphorylation of Cdk1 substrates. This observation is reminiscent of our recent report of a latent stable steady state that can be reached in prophase (Rata *et al*, 2018). These early prophase substrates are likely to be poor PP2A:B55 targets, or highly efficient Cdk1 substrates, and therefore require a low threshold of Cdk1:cyclin activity. The next step in mitotic establishment, NEBD and transition to prometaphase, requires Greatwall-dependent PP2A:B55 inactivation, or cyclin B-dependent increase in the levels of Cdk1 activity to phosphorylate the bulk of intermediate mitotic substrates. We demonstrate that in the absence of cyclin B, Greatwall activation is unperturbed and this allows progression of cells to prometaphase.

This is likely to depend on Cdk1:cyclin A that can phosphorylate and activate Greatwall kinase *in vitro* (Mochida *et al*, 2016). Our finding that cyclin B and Greatwall depletion has additive effects suggests that they act in parallel, downstream of cyclin A to promote mitotic phosphorylation. This explains how this regulatory network is designed to allow buffering against fluctuations in kinase and phosphatase activity during mitotic entry. It also explains the lack of entry phenotypes in Greatwall and cyclin B-depleted cells. It is noteworthy that mitotic entry in embryonic cell cycles is much more reliant on both cyclin B and Greatwall (Castilho *et al*, 2009; Strauss *et al*, 2018). A possible explanation for this difference could be elevated levels of PP2A:B55 in these cells that have previously remained in prolonged G2 arrests in form of immature oocytes.

Our study defines a final set of late substrates that strictly require cyclin B-dependent Cdk1 activity. These phosphorylation events are critical to initiate anaphase and cell division. The cyclin B-dependent phospho-sites could be highly sensitive to Cdk1 counteracting phosphatases or poor Cdk1 substrates and therefore require higher levels of Cdk1:cyclin activity. Alternatively, defined localisation of cyclin B could also contribute to the observed substrate specificity and also further re-enforce localised Greatwall-dependent PP2A: B55. The importance of cyclin B localisation is supported by recent work (Alfonso-Pérez *et al*, 2019) showing a requirement for cyclin B1 to phosphorylate Mps1 at the kinetochore and promote the SAC. Our cyclin B1[dd]/B2[ko] cells will be a useful tool to further analyse the requirements for cyclin B in SAC signalling. Likewise, future work will need to establish the precise S-phase functions of cyclin A and its contributions to initiating mitosis. The rapid induced degradation system that we report here will be a helpful tool to further address these questions for a better understanding of mammalian cell cycle control mechanisms.

## Material and Methods

### Tissue culture and chemical reagents

hTERT RPE-1 cells were obtained from ATCC (cat. CRL-4000) and were grown at 37°C with 5% $CO_2$ in DMEM/F12 (Sigma-Aldrich) media containing 10% foetal bovine serum (FCS) and 1% penicillin–streptomycin. All cell lines were maintained by the GDSC cell culture facility, regularly authenticated and tested for mycoplasma infection.

Drugs used for this study and working concentrations were as follows:

PD-166285 (0.5 μM; Stratech Scientific Limited, S8148-SEL).
Apcin (26 μM; Sigma-Aldrich SML1503).
ProTAME (6 μM; Bio-techne I-440-01M).
Asunaprevir (3 μM; BMS-650032, Bioquote, A3195).
Indole-3-acetic acid (IAA, 500 μM; Sigma-Aldrich I5148).
Doxycycline (1 μg/ml; Takara-bio, Clontech 631311).
Thymidine (4 mM; Sigma-Aldrich T1895).
SiR-DNA (50–100 nM; Tebu-bio Ltd. SC007).
SiR-Tubulin (50–100 nM; Tebu-bio Ltd. SC002).

### Antibodies

A list of antibodies used in this study is provided in Appendix Table S1.

### Generation of endogenously tagged cell lines

gRNAs were designed using Benchling CRISPR tool (https://benchling.com/). The sequences 5′ ACTAGTTCAAGATTTAGCCA 3′, 5′ TGTTTCTAAAACCATCAAGT 3′, 5′ GCACACTCACCGTCGGGCGT 3′ and 5′ GACCTGCTACAGGCACTCGT 3′ were chosen for cyclin B1, cyclin A2, cyclin B2 and Rosa26, respectively, and introduced into the vector pSpCas9(BB)-2A-Puro (PX459) V2.0 following the protocol described in Ran *et al* (2013). PX459 acquired from Feng Zhang via Addgene (plasmid # 48139). Indel mutations in cyclin B2 were confirmed by Sanger sequencing as two frameshift mutations downstream of the initiating ATG in the CCNB2 gene (CTCGACG-C CCGACG-GTGAG and CTCGACGCC-C-GACGGTGAG with the missing residues marked by hyphenation). The puromycin resistance in hTERT RPE-1/OsTIR1 cells was removed using CRISPR using the following gRNA sequence: 5′ AGGGTAGTCGGCGAACGCGG 3′. To make the targeting template, Gibson assembly was used to assemble into NotI-digested pAAV-CMV vector (gift from Stephan Geley, University of Innsbruck, Austria) the fragments in the following order: the left arm, a linker (5′ CGCCTCAGCGGCATCAGCTGCAGGAGCTGG AGGTGCATCTGGCTCAGCGGCAGG 3′), mAID 3, SMASh 5, T2A-neomycin and the right arm. To get CRISPR-resistant constructs, the following sequences were mutated as followed: ACTAGTTCAAGATT TAGCCAAGG by AtTAGTcCAgGAccTAGCtAAaG for cyclin B1 and CCATCAAGTCGGTCAGACAGAAA by CCATgAtGaCGcTCAcACAGttA for cyclin A2. Mutations (lowercase letters) are silent and preferential codon usage was taken into account. For inducible expression of OsTIR1, we used the construct described in Natsume *et al* (2016), combined it with a bleomycin/zeocin resistance marker and cloned it into a Rosa26 targeting construct. Integration was confirmed by genomic PCR (Fig 1B and C). To generate stable clones, $10^6$ hTERT immortalised RPE-1 cells were transfected with 0.5 μg of gRNA/Cas9 expression plasmid and 1.5 μg of targeting template using Neon transfection system (Invitrogen), with the following settings: 10-μl needle, 1,350 V, 20 ms and two pulses. Clones were incubated for 3 weeks in media containing 1 mg/ml of neomycin (Sigma-Aldrich), 5 μg/ml blasticidin (Gibco) or 500 μg/ml zeocin (Invivogen) and selected clones were screened by Western blot.

### Generation of PCNA-tagged cell lines

AAV-293T cells (Clontech) were seeded into a T75 flask 1 day before transfection, such that they were 70% confluent on the day of transfection. Cells were transfected with 3 μg each of pAAV-mRuby-PCNA (Zerjatke *et al*, 2017) pRC and pHelper plasmids, and 20 μl Lipofectamine 2000, diluted in 3 ml OPTIMEM (Gibco). Lipofectamine/DNA mixture was added to cells in 7 ml of complete medium (DMEM with 10% FBS and 1% penicillin–streptomycin). Cells were incubated at 37°C for 6 h before medium was replaced with 10 ml of complete medium. Three days post-transfection, medium and cells were transferred to a 50-ml falcon tube. Cells were lysed with three rounds of freeze–thaw. The sample was centrifuged at 10,000 *g* for 30 min at 4°C. Supernatant containing AAV particles was collected and either used immediately or aliquoted and stored at −80°C. cyclin A2[dd] cells were plated 1 day before transduction, such that they were 40% confluent for transduction. Cells were washed twice in PBS and incubated in 5 ml of complete medium plus 5 ml of AAV-mRuby-PCNA containing supernatant for 48 h. Cells were expanded for a further 48 h followed by FACS sorting using a BD FACSMelody sorter according to the manufacturer's instructions.

### Generation of cell lines stably expressing fluorescent protein markers

For rapid generation of multiple fluorescent protein-tagged cellular markers, we cloned a sequence of P2A-ScaI-mEmeraldT2A-Balsticidin resistance marker into the pFusionRed-H2B expression construct

(Evrogen, FP421). The ScaI site was then used to clone Mis12 and AurB in-frame with the preceding P2A and the following T2A sequence. Cyclin A2$^{dd}$ and B1$^{dd}$B2$^{ko}$ cells were transfected with 2 μg of the expression plasmids by NEON electroporation (Invitrogen) and grown for 2 weeks in medium containing 5 μg/ml blasticidin (Gibco). Fluorescent protein expressing cell lines were isolated by FACS sorting using a BD FACSMelody sorter according to the manufacturer's instruction.

## Generation of sleeping beauty cell lines

TET-on Sleeping beauty plasmid was obtained from Addgene (plasmid nr. 60496 pSB-tet-BP) with a blue fluorescent protein (BFP) selection marker.

The plasmid originally contains Luciferase which was replaced by the ORF of cyclin B-YFP and cyclin B-YFP-NLS fusions using NEB HiFi Assembly. We used BspDI and NcoI sites to cut out the luciferase and incorporated our GOI. 1.9 μg of this plasmid along with 100 ng transposase enzyme SB-100X (Addgene plasmid nr. 34879) was transfected into RPE-1 degron cells using electroporation. Afterwards, cells were grown for 10 days and FACS sorted into a 96-well plate for BFP expression (excitation ~ 456 nm) using FACSMelody sorter according to the manufacturer's instructions. Cells were then grown up and analysed for protein expression after doxycycline addition using immunoblotting.

## Genomic PCR

Genomic DNA was extracted using DNeasy Blood and Tissue Kit (Qiagen) according to the instructor's recommendation; then, DNA was amplified with Phusion High Fidelity DNA polymerase (New England Biolabs) using the following primer pairs to check genomic integration: 5′ CTGCATTCTAGTTGTGGTTTGTCCA 3′ and 5′ ACT ATGACCCACGCAGTACAA 3′ (TetON-OsTIR1 into Rosa26 locus), 5′ ATTGCTGAAGAGCTTGGCGG 3′ and 5′ TCACACCATTCAAGCACC TGTA 3′ (degron tags into cyclin A2 locus), 5′ CTGAGCGGAAAACC TGCTATC 3′ and 5′ CTGAACGAACAGGGGAAATGGTT 3′ (degron tags into cyclin B1 locus), 5′ TGGTGGAAGATTGGTGGCTC 3′ and 5′ CTGCTTCTGGCATGGCTTTC 3′ (internal amplification control in Kif23 locus). Samples were loaded onto 1% agarose gel (Fisher) in 1× TAE buffer and containing ethidium bromide then imaged using Ingenius, Syngene Bio-imaging apparatus.

## RT–qPCR

cDNA was prepared from extracted mRNA (RNeasy kit, Qiagen) using oligo d(T) (Ambion), murine RNase inhibitor (New England Biolabs) and M-MuLV reverse transcriptase (New England Biolabs) following the manufacturer's directions. Samples were then treated with RNase A (Sigma) and cleaned using QIAquick PCR purification kit (Qiagen). Next, qPCR was carried out with HOT FIREPol EvaGreen qPCR Mix Plus (no Rox, Solis BioDyne) and primers for cyclin B3 (5′ AAGACACTGACCTTGTCCCG 3′ and 5′ AGAGGGC-CAGGAGTAAGGAG 3′) and the loading controls: actin (5′ GAAGTG TGACGTGGACATCC 3′ and 5′ CTCGTCATACTCCTGCTTGC 3′) and TATA-binding protein (5′ CACGAACCACGGCACTGATT 3′ and 5′ TTTTCTTGCTGCCAGTCTGGAC 3′). The reaction was performed on a PCR Stratagene MX3005P system. Fold change in expression over

control was calculated using the delta-delta Ct method (Livak & Schmittgen, 2001).

## Proliferation assay

One thousand cells per well were seeded and treated or not with 1 μg/ml of doxycycline, 3 μM of Asv and 500 μM of IAA for 1 week; then, cells were fixed and stained with a solution containing 0.05% crystal violet (Sigma-Aldrich), 1% formaldehyde (Sigma-Aldrich), 1% methanol in PBS for 10 min.

## FACS analysis

Cells were incubated with 10 μM of EdU for 1 h before being harvested and fixed in 70% ethanol at 4°C overnight. Next, EdU was labelled with the fluorophore Alexa 647 using Click-iT EdU Imaging Kit (Invitrogen) and DNA were stained with 5 μg/ml of propidium iodide (Fluka) and 150 μg/ml of RNAseA (Sigma-Aldrich). FACS profiles were obtained from data acquired on Accuri C6 Flow cytometer (BD Biosciences).

## EdU pulse chase

Cells were incubated with or without the DIA cocktail for 4 h. Afterwards, 10 μM EdU was added for 1 h and washed 3–5× with PBS before re-incubating in media with or without the DIA cocktail. Starting at EdU washout, cells were harvested every 3 h for 12 h and samples prepared and analysed as noted previously.

## Cell synchronisation

To enrich mitotic cells and block cells in metaphase, we used a single 24-h Thymidine block release. Apcin (25 μM) and proTAME (6 μM) were added 12 h after the release for 2 h before fixation and analysis. For more efficient mitotic enrichment to collect cells for proteomic or immunoblotting analysis, we pre-synchronised cells by serum starvation for 72 h, followed by a 24-h release into serum and Thymidine containing medium, followed by release from Thymidine. ProTAME (6 μM) was added 12 h after release for 2–4 h and mitotic cells were collected by shake-off.

## siRNA transfection

siRNAs were resuspended to 20 μM stock concentration. 1.5 ml 20 × 10$^4$ cells were reverse transfected with siRNA diluted in 500 μl MEM (Gibco) media containing (10 μl) siRNA-MAX reagent (Invitrogen) and a final concentration of 80 nM siRNA. After 7 h-O/N incubation, the media was changed to the standard growth media. The cells were subsequently used for analysis after 48 h, or 72 h in the case of CCNB3. The following siRNAs were used for this study siGWL (Qiagen Ltd SI02653182 Hs*MASTL*7 FlexiTube) and "on-target plus" smart-pool siRNAs (Dharmacon/Horizon) for ARPP19, ENSA, CCNB3, CCNB1, CCNB2, CCNA2, CCNA1, CDC27, B55α, B55δ. As a negative control, we used All Stars negative control siRNA (Qiagen 1027280). In the case of Gwl and ENSA/ARPP siRNA transfections, cells were synchronised 12 h after siRNA transfection by a 25-h single Thymidine block and analysed 10–12 h after release.

## Immunoblotting

Cells were pre-treated for 2 h with 1 μg/ml of doxycycline (Clontech) and then 3 μM of asunaprevir (Asv; ApexBio) and/or 500 μM IAA were added to the media for indicated time periods. Cells were harvested and lysed in EBC buffer (50 mM Tris pH 7.5, 120 mM NaCl, 0.5% NP-40, 1 mM EDTA, 1 mM DTT, protease and phosphatase inhibitors [Complete and PhosStop; Roche Diagnostics]) then mixed with 5× sample buffer (0.01% bromophenol blue, 62.5 mM Tris–HCl pH 6.8, 7% SDS, 20% sucrose and 5% β-mercaptoethanol). The samples were sonicated then boiled at 95°C for 5 min. Samples were analysed by Western blotting and the signal was detected using Immobilon Western Chemiluminescent HRP substrate (Millipore). The intensity of cyclins A2 and B1 signals was quantified using ImageJ software. GAPDH was used to normalise the samples.

## Immunofluorescence

Cells were fixed for 10 min with 3.7% formaldehyde (Sigma-Aldrich) in PBS or with 100% ice-cold methanol for 1 min, washed in PBS and permeabilised with PBS-containing 0.5% NP-40 for 10–20 min. Following 30 min blocking in 3% bovine serum albumin (BSA, Sigma-Aldrich), cells were labelled with the indicated antibodies diluted in 3% BSA/PBS. Secondary antibodies were labelled with Alexa Fluor dyes purchased from Invitrogen. Cell nuclei were counterstained with 4,6-diamidino-2-phenylindole dihydrochloride (DAPI; Sigma-Aldrich) then slides were mounted with ProLong Diamond Antifade Mountant (Invitrogen), or if cells were grown in 96-well PerkinElmer CellCarrier Ultra polystyrene plates (PerkinElmer cat. no 6005550), they were left in 300 μl PBS at 4°C and analysed within 1 week.

## Cold treatment

Cells were grown on coverslips and were or not pre-treated with 1 μg/ml of doxycycline for 2 h; then, 3 μM of Asv and 500 μM of IAA were added for 4 h. Next, the cells were incubated on ice for 10 min. For microtubule regrowth assay, the cells were re-incubated at 37°C for indicated times. Fixation was carried out with PHEM (60 mM Pipes, 25 mM HepesKOH pH 7.0, 5 mM EGTA, 4 mM MgSO4, 0.5% NP-40 and 3.7% formaldehyde) for 5 min followed by 95% ice-cold methanol/5 mM EGTA for 5 min. The samples were then subjected to immunofluorescence, widefield microscopy and deconvolution.

## Chromosome spreads

Cells were synchronised by single Thymidine release and proTAME/Apcin arrest as described above. Mitotic cells were collected by shake-off and resuspended in warm 75 mM KCl solution for 10 min before centrifugation. After aspirating KCl, methanol: acetic acid (3:1) was added to cells and incubated at RT for 5 min. Cells were centrifuged and resuspended in 300–500 μl remaining solution and dropped on glass coverslips. They were allowed to dry for 1 h prior to additional 4% Formaldehyde fixation for 10 min followed by immunofluorescence using Ki67 antibodies.

## Widefield live-cell microscopy

Cells were seeded on μ-slides from Ibidi. Cells were pre-treated with 2 μg/ml of doxycycline for 2 h before imaging, and 3 μM of Asv and 500 μM of IAA were added at the beginning of the imaging. To observe the nucleus, cells were pre-treated with 50 nM SiR-DNA (Spirochrome) for 2 h. To analyse mitotic entry, time-lapse microscopy was performed in an environmental chamber (Digital Pixels, Microscopy Systems & Solutions) heated at 37°C with 5% $CO_2$ supply using an Olympus IX71 equipped with Orca-flash4.0LT camera and a LUCPlanFLN, NA 0.45, 20× objective, or 0.64 NA, LUCPlanFLN 40× lens and 2 × 2 binning. Cells were imaged using differential interference contrast (DIC), and fluorescent illumination using a Lumencor Spectra LED light source, and 640/40 excitation, 705/72 emission filters. Images were acquired every 5 min using Micro-Manager v1.4 software and cells that rounded-up and condensed their nucleus were scored.

## Widefield immunofluorescence microscopy and deconvolution

Slides were imaged using an Olympus IX70 equipped with CoolSNAP HQ2 camera and an UApo N 340 NA 1.35, 40× oil immersion objective controlled by Micro-Manager v1.4 software. Stacks were taken at 0.2-μm intervals and deconvolved using the Huygens classic maximum-likelihood estimation algorithm (Scientific Volume Imaging) based on a measured point spread function.

## Airyscan confocal microscopy (Zeiss)

Confocal imaging was performed on the Airy scan module of Zeiss LSM880 (Carl Zeiss AG®, Germany) with Plan-Apochromat 63x/1.4 Oil objective in a live support chamber (Digital Pixels, Microscopy Systems & Solutions) at 5% $CO_2$ and 37°C. The "Fast" scanning mode was used to speed up the imaging. At each imaging position, seven axial slices with 2-μm interval were taken in three fluorescence channels including EGFP, RFP and Cy5. The excitation wavelengths were 488, 561 and 633 nm. The emission filters in front of the Airyscan detectors were BP495-550, BP495-620 and LP645, respectively. Each image was accumulated from four scans to minimise laser-induced photo-bleaching and photo-toxicity. The time lapses were set to last over 18 h with 5-min intervals. Raw images were processed using Airyscan processing algorithm in Zen (Carl Zeiss AG®, Germany). The processed image sequence was then maximum projected.

## Spinning disc confocal imaging on operetta (PerkinElmer)

High-throughput imaging was performed on Operetta CLS (PerkinElmer Ltd@) in confocal mode with 40×/0.9 water objective. Cells were cultured and imaged in PerkinElmer CellCarrier Ultra 96-well plates. At each imaging point, three axial slices with 4-μm interval were scanned every 5 min in three fluorescence channels including EGFP, RFP and Cy5. The imaging lasted around 18 h in 5% $CO_2$ and 37°C live environment. The image sequence was maximum projected and then analysed using Harmony software.

**Image segmentation and quantification of immunofluorescence**

Regions of interest were either manually generated in ImageJ or using Harmony segmentation algorithms. Mean and sum intensity were estimated in ImageJ or Harmony and plotted using Python Pandas, Matplotlib and Seaborn APIs (https://www.python.org, https://matplotlib.org,https://seaborn.pydata.org).

**High pH reverse-phase chromatography and phospho-enrichment**

Control and DIA-treated cells were lysed in 2% SDS in Dulbecco's PBS-containing protease and phosphatase inhibitors (Roche; mini-cOmplete protease inhibitors EDTA-free, PhosStop). Lysates were homogenised by sonication using a probe sonicator (Branson soni-fier, 20% power, 30 s). Two hundred microgram protein was reduced with 25 mM TCEP (Thermo Pierce), alkylated with *N*-ethylmaleimide (Sigma, 55 mM prepared fresh) and then precipitated using the chloroform–methanol method. The protein precipitate was resuspended in 0.1 M triethylammonium bicarbonate (TEAB), pH 8.5 and digested first with 2 µg LysC (Wako) for 4 h and then 2 µg trypsin (Thermo Pierce) overnight. The digest was then reacted with 6-plex TMT reagents (Thermo Pierce) using the manufacturer's recommended protocol. The TMT-labelled peptides were then mixed.

Peptides were separated off-line by high pH reverse-phase chromatography using an Ultimate 3000 high-performance/pressure liquid chromatography system (HPLC; Thermo Dionex) equipped with a BEH 4.6 cm × 150 mm BEH C18 column (Waters) using the following mobile phases: (A) 10 mM ammonium formate, pH 9.3 and (B) 10 mM ammonium formate, pH 9.3 in 80% acetonitrile (ACN). Peptides were eluted using a linear gradient into 24 fractions. Five percent of each fraction was dried and retained aside for total proteome analysis. The remaining 95% was taken forward for phospho-enrichment.

Phospho-enrichment was performed using magnetic Ti:IMAC beads according to the manufacturer's instructions (Resyn Biosciences), similar to previously described (Hiraga *et al*, 2017). Dried fractions were resuspended in load buffer (80% ACN, 5% TFA, 5% glycolic acid) and combined into 16 fractions in the following manner: F17 + F1, F18 + F10, F19 + F11, F20 + F12, F21 + F14, F23 + F15, F24 + F16. Enriched phosphopeptides were concentrated by evaporation.

**LC-MS/MS and MS data analysis**

Peptides were resuspended in 5% formic acid and separated by reverse-phase chromatography on a Ultimate3000 RSLCnano HPLC (Thermo Dionex) equipped with an EasySpray 75 µm × 50 cm column (Thermo Scientific). Peptides were eluted using a linear elution gradient from 2 to 35% B over 2 h using the following mobile phases: (A) 0.1% formic acid and (B) 80% ACN + 0.1% formic acid. Peptides were then ionised and desolvated in an EasySpray electrospray source with a 2.5 kV voltage applied between the emitter and the ion transfer capillary front-end of a Fusion Linear Ion Trap-Orbitrap mass spectrometer (Thermo Scientific). The acquisitions were data-dependent. For MS1 scans, ions from 350 to 1,400 $m/z$ were selected using the quadrupole and $m/z$ scanned using the Orbitrap at 120,000 resolution. For MS2 scans, the instrument was operated in top speed mode. Precursors were selected using a 1.6 $m/z$ window for CID fragmentation at 30 normalised collision energy units followed by $m/z$ measurement in the linear ion trap. The top 3 MS2 fragments were then selected for synchronous precursor selection (SPS) using an isolation window of 2 $m/z$ and a maximum injection time of 300 ms. The MS2 fragments were then subjected to further HCD fragmentation at 55 normalised collision energy units. An MS3 scan of the HCD fragments was then performed in the Orbitrap at 60,000 resolution using a scan range of 100 to 150 $m/z$.

Peptide identification and quantitation were performed using MaxQuant (version 1.5.6.5) (Cox & Mann, 2008), which incorporates the Andromeda search engine (Cox *et al*, 2011). Default parameters for TMT quantitation were used in MaxQuant, allowing for the following variable modifications: phospho(STY), protein N-terminal acetylation, glutamine to pyro(glutamate) conversion and deamidation of Gln and Asn. The human reference proteome from UniProt (accessed on 25 July 2016) was used as the sequence database for the identification. Raw Data are available via ProteomeXchange with identifier PXD012100. Post-MaxQuant processing was performed using R (version 3.5.0). Potential contaminant proteins and hits to the reverse database were removed from consideration. TMT reporter was normalised by the median protein TMT reporter intensities to correct for unequal loading and also corrected for isotope contamination. Phosphorylation site ratios were then calculated for each of the three biological replicates and normalised to the ratios measured for the total protein. Fold change cut-offs at $\bar{x} \pm 1.96s$ was determined by modelling log ratios as a normal distribution. Motif analysis was performed by mapping regular expressions downloaded for phosphorylation consensus motifs from ELM. This was performed on a filtered dataset considering only those phosphorylation sites with a localisation probability above 0.75.

**Functional classification of the candidates**

Outliers were analysed for biological process ontology. Annotations were extracted from the Gene Ontology database using QuickGO. Only qualifier ECO:0000269 experimental evidence used in manual assertion was considered. Biological process ontology verified by direct assay or mutant phenotype from Uniprot database was used to complement the protein classification.

**Protein interaction map**

The interactome was built from human protein–protein interaction (PPI) databases. All outliers were considered to build a PPI map using the software Cytoscape 3.5.1. Human PPIs from databases were retrieved using the plugin Bisogenet. Small-scale studies were considered and *in vivo* interactions or PPIs from direct complexes were displayed. The PPI map was complemented using data concerning protein complexes obtained from Uniprot and Corum (http://mips.helmholtz-muenchen.de/corum/). Only PPIs relevant or possibly related to mitosis and proteins present in several protein complexes were shown.

**Mathematical modelling of mitotic substrate phosphorylation**

We have extended our previous model (Rata *et al*, 2018) with three different (early, intermediate and late) mitotic substrates showing

different sensitivities to cyclin:Cdk1 complexes and PP2A:B55 phosphatase (see Fig 9). CycB:Cdk1, but not CycA.Cdk1, is regulated by inhibitory Cdk1-phosphorylation controlled by Wee1 and Cdc25. Both of the Tyr-modifying enzymes are regulated by CycB:Cdk1 itself as well as by CycA:Cdk1 and PP2A:B55. The activity of PP2A: B55 is controlled by the Greatwall-ENSA pathway which is activated by cyclin:Cdk1-dependent phosphorylation of Greatwall kinase. The early mitotic (prophase) substrates could be phosphorylated by both cyclin:Cdk1 complexes and dephosphorylated by an unknown phosphatase. In contrast, the intermediate (prometaphase) and the late mitotic substrates are both dephosphorylated by PP2A:B55. While the intermediate substrates are phosphorylated by both cyclin:Cdk1 complexes, the phosphorylation of late substrates requires CycB: Cdk1 activity. We used nonlinear ordinary differential equations to describe the rate of change for protein concentrations in the network.

All the reactions are described by law of mass action kinetics except for the early mitotic substrates (pSearly), Wee1 and Cdc25 which are approximated by the Hill-kinetics. The rate constants are abbreviated by "k" with subscripts referring to the type of reaction (a-activation, i-inhibition, p-phosphorylation and dp-dephosphorylation) and the substrate of the reaction. The numerical values of kinetic parameters are provided in the XPPAut "ode" code that allows the users to reproduce Fig 9 (see Code in EV1).

### Statistical analysis

All experiments included at least three independent biological repeats. Sample size per repeat varied between experiments and is indicated in the figure legends. Sample size was based on standard practice in cell biological assays and not specifically pre-estimated. *P*-values were calculated using an independent two-sample t-test. Levels of significance are indicated by stars (*$P < 0.05$, **$P < 0.01$, ***$P < 0.001$). For all experiments, samples were not randomised and the investigators were not blinded to the group allocation during experiments and outcome assessment. No exclusion criteria were used and all collected data were used for statistical analysis.

## Data availability

Cell lines and reagents to establish endogenous double degron tags are available upon request (contact HH at hh65@sussex.ac.uk). The mass spectrometry data from this publication have been deposited to the ProteomeXchange partner PRIDE database (https://www.ebl.ac.uk/pride/) and assigned the identifier PXD012100 (http://www.ebi.ac.uk/pride/archive/projects/PXD012100).

**Expanded View** for this article is available online.

### Acknowledgements
We thank the members of the Hochegger, Ly, Lamond and Novak laboratories for supporting work on this study. We thank Tim Hunt and Randy Poon for critical evaluation of this MS and Viji Draviam for discussing results and sharing reagents. HH was a CRUK senior research fellow (C28206/A14499), and TL is supported by a Sir Henry Dale Fellowship jointly funded by the Wellcome Trust and the Royal Society (ID 206211/Z/17/Z). HH and BN were supported by a BBSRC LoLa grant (ID BB/M00354X/1). We acknowledge the support of the Wolfson Foundation (Grant ref 20440) for funding the Zeiss LSM880 and Operetta CLS microscopes used in this study.

### Author contributions
TL and HH designed this study, performed experiments, analysed data and wrote the MS. NH, AC, MSPR, FEI and OB performed experiments and data analysis and helped with MS preparation. ARB and CB generated PCNA-tagged cell lines. YG supported the microscopy experiments. BN and PFL performed the mathematical modelling and helped prepare the MS. AIL supported the proteomic analysis. MTK provided unpublished reagents.

### Conflict of interest
The authors declare that they have no conflict of interest.

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
