## [Review Process File · The EMBO Journal]

Cyclin A triggers Mitosis either via the Greatwall kinase pathway or Cyclin B

Helfrid Hochegger, Bela Novak, Masato Kanemaki, Chris Bakal, Angus I. Lamond, Tony Ly, Adrijana Crncec, Nadia Hegarat, Maria Suarez Pereda Rodriguez, Fabio Echegaray Iturra, Yan Gu, Oliver Busby, Paul Lang and Alexis Barr.

Review timeline:	Submission date:	7 th January 2020
	Editorial Decision:	21 st January 2020
	Revision received:	9 th March 2020
	Accepted:	25 th March 2020

Editor: Hartmut Vodermaier

Transaction Report:

Please note that this manuscript was previously reviewed at another journal. Since the original reviews are not subject to *The EMBO Journal's* transparent review process policy, these initial reports and author response to them cannot be published here.

1st Editorial Decision

21st January 2020

Thank you again for submitting/transferring your manuscript together with previous reports and responses from another journal for our editorial consideration. As discussed earlier, we have now consulted with a trusted arbitrating referee of our own journal, who had access to both the latest (revised) manuscript and to the original comments and your response to them. I am pleased to say that our arbitrator (see comments below) considered the study interesting and the key issues raised by the original referees adequately addressed, only asking for various corrections and restructuring of the manuscript. Following this, we shall therefore be happy to accept the manuscript for EMBO Journal publication.

REFeree REPORTS

Referee #1:

This is an interesting and provocative study of the relative roles of cyclins A and B in human mitosis. Numerous previous papers have provided evidence for the different functions of the two cyclins, but the current paper addresses this problem more effectively by using a clever new strategy to acutely deplete the cyclins. I have no major concerns and I believe this work is suitable for publication in EMBO Journal.

I have only minor comments:

1. I can agree with the previous reviewer 1 that the paper is quite dense and the experimental details

are not always clear in the main text or in the figure legends. I understand that the paper was written for a short format in its previous submission, but I believe it would benefit from some unpacking for EMBO Journal.

2. I found a few errors:

- No legend text is provided for Fig 3I-K.
- The labeling of 3I seems incorrect: there should be a plus sign above lane 5.
- in Fig 4B, the percentage is cut off for cytoskeleton.
- The dot plots in 4A, C, and H are terrible quality due to compression artifacts, etc.
- The number of proline-directed sites is unclear: is it 3192 (page 9, para 2, line 6) or is it 2351 (line 10)?

1st Revision - authors' response

9th March 2020

Response to Referee Reports.

Referee #1:

This is an interesting and provocative study of the relative roles of cyclins A and B in human mitosis. Numerous previous papers have provided evidence for the different functions of the two cyclins, but the current paper addresses this problem more effectively by using a clever new strategy to acutely deplete the cyclins. I have no major concerns and I believe this work is suitable for publication in EMBO Journal.

I have only minor comments:

1. I can agree with the previous reviewer 1 that the paper is quite dense and the experimental details are not always clear in the main text or in the figure legends. I understand that the paper was written for a short format in its previous submission, but I believe it would benefit from some unpacking for EMBO Journal.

We have reformatted the MS to make the data more accessible.

2. I found a few errors:

- No legend text is provided for Fig 3I-K.
- The labeling of 3I seems incorrect: there should be a plus sign above lane 5.
- in Fig 4B, the percentage is cut off for cytoskeleton.
- The dot plots in 4A, C, and H are terrible quality due to compression artifacts, etc.
- The number of proline-directed sites is unclear: is it 3192 (page 9, para 2, line 6) or is it 2351 (line 10)?

We would like to thank the reviewer for their careful assessment. We have addressed the errors pointed out above.

Accepted

25th March 2020

Thank you for submitting your final revised manuscript for our consideration. I am pleased to inform you that we have now accepted it for publication in The EMBO Journal.

Corresponding Author Name: Helfrid Hochegger

Journal Submitted to: EMBOJ

Manuscript Number: 104419